# 🕷 MODEL SPIDER: Learning to Rank Pre-Trained Models Efficiently

**Yi-Kai Zhang**[1], **Ting-Ji Huang**[1], **Yao-Xiang Ding**[2], **De-Chuan Zhan**[1], **Han-Jia Ye**[1,✉]

[1]National Key Laboratory for Novel Software Technology, Nanjing University, China
[2]State Key Lab of CAD & CG, Zhejiang University
`{zhangyk,huangtj,zhandc,yehj}@lamda.nju.edu.cn  yxding@zju.edu.cn`

## Abstract

Figuring out which Pre-Trained Model (PTM) from a model zoo fits the target task is essential to take advantage of plentiful model resources. With the availability of numerous heterogeneous PTMs from diverse fields, *efficiently* selecting the most suitable one is challenging due to the time-consuming costs of carrying out forward or backward passes over all PTMs. In this paper, we propose MODEL SPIDER, which *tokenizes* both PTMs and tasks by summarizing their characteristics into vectors to enable efficient PTM selection. By leveraging the approximated performance of PTMs on a separate set of training tasks, MODEL SPIDER learns to construct representation and measure the fitness score between a model-task pair via their representation. The ability to rank relevant PTMs higher than others generalizes to new tasks. With the top-ranked PTM candidates, we further learn to enrich task repr. with their PTM-specific semantics to re-rank the PTMs for better selection. MODEL SPIDER *balances efficiency and selection ability*, making PTM selection like a spider preying on a web. MODEL SPIDER exhibits promising performance across diverse model zoos, including visual models and Large Language Models (LLMs). Code is available at `https://github.com/zhangyikaii/Model-Spider`.

## 1 Introduction

Fine-tuning Pre-Trained Models (PTMs) on downstream tasks has shown remarkable improvements in various fields [35, 26, 75, 42, 16], making "pre-training → fine-tuning" the de-facto paradigm in many real-world applications. A model zoo contains diverse PTMs in their architectures and functionalities [1, 12], but a randomly selected PTM makes their helpfulness for a particular downstream task vary unpredictably [80, 70, 102]. One important step to take advantage of PTM resources is to identify the most helpful PTM in a model zoo — estimating and ranking the transferabilities of PTMs — with the downstream task's data *accurately and efficiently*.

Which PTM is the most helpful? A direct answer is to enumerate all PTMs and evaluate the performance of their corresponding fine-tuned models. However, the high computational cost of the backward steps in fine-tuning makes this solution impractical. Some existing methods estimate proxies of transferability with only forward passes based on the target task's features extracted by PTMs [9, 97, 66, 55, 113, 27, 71, 25, 93]. Nowadays, a public model zoo often contains hundreds and thousands of PTMs [104]. Then, the computational burden of forward passes will be amplified, let alone for the time-consuming forward passes of some complicated PTMs. Therefore, the *efficiency* of searching helpful PTMs and estimating the transferability should be further emphasized.

In this paper, we propose MODEL SPIDER, the SPecification InDuced Expression and Ranking of PTMs, for accurate and efficient PTM selection. In detail, we tokenize all PTMs and tasks into vectors that capture their *general properties* and their relationship with each other. For example,

37th Conference on Neural Information Processing Systems (NeurIPS 2023).

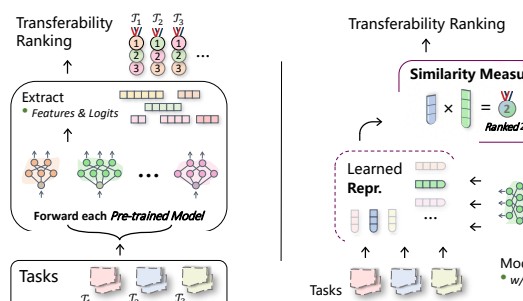
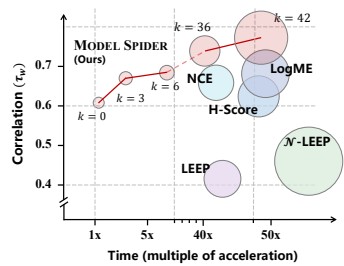

| (a) **Forward**-Based and **Representation/Specification**-Based Model Selection | (b) Selection ability & Efficiency Comparison |

Figure 1: (a) Two strategies for PTM selection. Related works utilize forward-based features and corresponding proxies on the target dataset to evaluate transferability. The representation/specification-based approach with learned model-task pair reduces the requirement for forwarding pass on each PTM. (b) The average efficiency (wall-clock time) *vs* performance (correlation $\tau_w$, the higher, the better) comparison of PTM selection. The circle sizes indicate the memory footprint. Red circles are MODEL SPIDER with different values of the number of PTM-specific features $k$, while others are comparison methods. MODEL SPIDER *balances efficiency and accuracy well*.

two models pre-trained on NABirds [37] and Caltech-UCSD Birds [100] datasets may have similar abilities in bird recognition. The comprehension abilities of two models pre-trained on XSum [64] dataset, Ax-b, and Ax-g datasets of SuperGLUE benchmark [101] may also be mutually transferable. We can then associate them with similar representation. Then the transferability from a PTM to a task could be approximated by the distance of their repr. *without requiring per-PTM forward pass over the downstream task*. The success of MODEL SPIDER depends on two key factors. First, how do we obtain representation for tasks and PTMs? The representation of the most helpful PTM should be close to the task one w.r.t. some similarity measures. Then, will a general task repr. weaken the selection ability since it may ignore specific characteristics of a PTM?

In MODEL SPIDER, we *learn* to construct representation with a general encoder and measure the similarity between them with a Transformer module [98] in a *supervised learning manner*. We estimate the rankings of PTMs in the model zoo for some historical tasks using rank aggregation. By leveraging the approximated supervision, we pull task representation close to the top-ranked PTM repr. and push unhelpful PTM repr. away based on the transformer-measured similarity. We expect that the ability to tokenize and measure similarity could be generalized to unseen tasks. The difference between MODEL SPIDER's representation-based PTM selection with forward-based strategy is illustrated in Figure 1.

The representation generated by general encoders significantly reduces the PTM search time and improves the search performance. If the budget allows, we can extract features of the downstream task by carrying out forward passes over *a part of* (the top-$k$ ranked) PTMs, revealing the *specific* relationship between PTMs and the task. We equip our MODEL SPIDER with the ability to incorporate PTM-specific representation, which re-ranks the PTMs and further improves the selection results. In summary, MODEL SPIDER is suitable for different budget requirements, where the general and task-specific repr. makes a flexible trade-off between efficiency and accuracy, given various forward passes. Figure 1 illustrates a comparison of PTM selection methods *w.r.t.* both efficiency and accuracy. Our contributions are

- We propose a novel approach MODEL SPIDER to tokenize tasks and PTMs, which is able to rank PTMs in a model zoo given a downstream task efficiently and accurately.
- MODEL SPIDER learns to tokenize and rank PTMs on a separate training set of tasks, and it can incorporate task-specific forward results of some PTMs when resource budgets allow.
- The experiments demonstrate that MODEL SPIDER effectively ranks PTMs and achieves significant improvements on the visual models and the Large Language Models (LLMs).

## 2 Related Works

**Efficient PTM Search with Transferability Assessment.** Whether a selected PTM is helpful could be formulated as the problem measuring transferability from source data pre-training the PTM to the target downstream task [111, 13, 41, 4, 78]. The current evaluation of transferability relies on a

forward pass of the PTM on the target task, which generates the PTM-specific features on the target task. For example, NCE [97], LEEP [66], LogME [113, 114], PACTran [27], and TransRate [39] estimate negative conditional entropy, log expectation, marginalized likelihood, PAC-Bayesian bound, mutual information to obtain proxy metric of transferability, respectively. Several extensions including $\mathcal{N}$-LEEP [55] with Gaussian mixture model on top of PTM features, H-Score [9] utilizing divergence transition matrix to approximate the transferred log-likelihood, and [25, 71, 84] exploring correlations between categories of target task. Auxiliary information such as source clues [6, 93] and gradients of PTMs when back propagating with few steps [85, 74] are also investigated. Although the transferability assessment methods avoid the time-consuming fine-tuning, the forward costs over PTMs also become heavier given diverse and complicated pre-trained model zoos.

**Relatedness of Task**. Whether a PTM gains improvements after fine-tuning on the downstream task has been verified to depend on the relatedness between tasks both theoretically [10, 11, 60] and empirically [102, 58, 112]. The relatedness could be measured through various ways, such as fully fine-tuning [115], task vectors [2], example-based graphs [48, 29, 86], representation-level similarities [30, 3], and human prior knowledge [44, 76]. Instead of utilizing a pre-defined strategy to measure the relatedness, MODEL SPIDER constructs the representation of PTMs/tasks in vector forms and learns a similarity between them on historical tasks.

**Learning to rank** predicts the orders of objects usually with a score function [43], and the experience on a training set could be generalized to unseen data [5, 63]. Additional learned metrics or embeddings further improve the ranking ability [62, 110, 15]. The task relatedness can also be modeled as a learning-to-rank problem, where the preference over one PTM over another could be learned from historical rankings of PTMs. However, obtaining the supervision on the training set requires complete fine-tuning over a large number of historical tasks, which either come from a time-consuming transfer learning experience [103] or the output from some specially selected transferability assessment methods [28]. We propose a strong and efficient approximation of the PTM ranking supervision on the training set tasks, and a novel representation-based similarity is applied.

# 3 Preliminary

We describe the PTM selection problem by assuming all PTMs are classifiers, and the description could be easily extended to PTMs for other tasks, *e.g.*, regression. Then we discuss several solutions.

## 3.1 Selecting PTMs from a Model Zoo

Consider we have a target classification task $\mathcal{T} = \{(\boldsymbol{x}_i, y_i)\}_{i=1}^{N}$ with $N$ labeled examples, where the label $y_i$ of each instance $\boldsymbol{x}_i$ comes from one of the $C_{\mathcal{T}}$ classes. Instead of learning on $\mathcal{T}$ directly, we assume there is a model zoo $\mathcal{M} = \{f_m = \boldsymbol{W}_m \circ \boldsymbol{\phi}_m\}_{m=1}^{M}$ containing $M$ PTMs. A PTM $f_m$ could be decomposed into two components. $\boldsymbol{\phi}_m$ is the feature extraction network producing $d_m$-dimensional features. $\boldsymbol{W}_m \in \mathbb{R}^{d_m \times C_m}$ is the top-layer classifier which maps a $d_m$-dimensional feature to the confidence score over $C_m$ classes.[1] PTMs in $\mathcal{M}$ are trained on source data across various domains. Their feature extractors $\boldsymbol{\phi}_m$ have diverse architectures, and the corresponding classifiers are pre-trained for different sets of objects. In other words, $d_m$ and $C_{m'}$ may differ for a certain pair of $m$ and $m'$. A widely-used way to take advantage of a PTM $f_m = \boldsymbol{W}_m \circ \boldsymbol{\phi}_m$ in the target task is to fine-tune the feature extractor together with a randomly initialized classifier over $\mathcal{T}$. In detail, we minimize the following objective

$$\hat{f} = \hat{\boldsymbol{W}} \circ \hat{\boldsymbol{\phi}} = \arg\min_{f=\boldsymbol{W}\circ\boldsymbol{\phi}} \sum_{i=1}^{N} \ell(\boldsymbol{W}^{\top}\boldsymbol{\phi}(\boldsymbol{x}_i), y_i \mid \boldsymbol{\phi}_m), \tag{1}$$

where $\boldsymbol{\phi}$ is *initialized with* $\boldsymbol{\phi}_m$. The fine-tuned $f$ makes prediction with $\arg\max_{c \in [C]} \hat{\boldsymbol{w}}_c^{\top} \hat{\boldsymbol{\phi}}(\boldsymbol{x})$. $[C] = \{1, \ldots, C\}$ and $\hat{\boldsymbol{w}}_c$ is the $c$th column of $\hat{\boldsymbol{W}}$. Then, we can rank the helpfulness of PTMs based on the performance of their fine-tuned models. In other words, we obtain $\hat{f}_m$ following Equation 1 based on the $m$th PTM $f_m$, then we calculate the averaged accuracy when predicting over an unseen test set of $\mathcal{T}$ (the higher, the better), *i.e.*,

$$\mathrm{t}_{\boldsymbol{\phi}_m \to \mathcal{T}} = \mathbb{E}\left[\mathbb{I}\left(y = \arg\max_{c \in [C]} \hat{f}_m(\boldsymbol{x})\right)\right]. \tag{2}$$

---

[1]We omit the bias term for simplicity.

$\mathrm{t}_{\phi_m \to \mathcal{T}}$ is also named as the *transferability*, measuring if the feature extractor $\phi_m$ in a PTM could be transferred well to the target task with fine-tuning [97, 39]. $\mathbb{I}(\cdot)$ is the indicator function, which outputs 1 if the condition is satisfied. Given $\boldsymbol{t}_\mathcal{T} = \{\mathrm{t}_{\phi_m \to \mathcal{T}}\}_{m=1}^M$, *i.e.*, the transferability for all PTMs, then we can obtain the ground-truth ranking of all PTMs in the model zoo for task $\mathcal{T}$ and select the top-ranked one. In the PTM selection problem, the goal is to estimate the ranking of all PTMs for a task $\mathcal{T}$ using $\hat{\boldsymbol{t}}_\mathcal{T} = \{\hat{\mathrm{t}}_{\phi_m \to \mathcal{T}}\}_{m=1}^M$. The evaluation criterion is the similarity between the predicted $\hat{\boldsymbol{t}}_\mathcal{T}$ and the ground-truth $\boldsymbol{t}_\mathcal{T}$, typically measured by weighted Kendall's $\tau_w$ [45]. We omit the subscript $\mathcal{T}$ when it is clear from the context.

## 3.2 Efficiency Matters in PTM Selection

One direct solution to PTM selection is approximating the ground truth $\boldsymbol{t}_\mathcal{T}$ by fine-tuning all the PTMs over $\mathcal{T}$, where a validation set should be split from $\mathcal{T}$ to estimate Equation 2. Since fine-tuning PTM contains multiple forward and backward passes, the computation burden is astronomical.

A forward pass of a certain PTM's extractor $\phi_m$ over $\mathcal{T}$ generates the features $\Phi_\mathcal{T}^m = \{\phi_m(\boldsymbol{x}_i) \in \mathbb{R}^{d_m}\}_{(\boldsymbol{x}_i, y_i) \in \mathcal{T}}$, which is lightweight compared with the backward step. The feature reveals how examples in $\mathcal{T}$ are distributed from the selected PTM's view, and a more discriminative feature may have a higher transfer potential. As mentioned in section 2, the existing transferability assessment methods estimate $\mathrm{t}_{\phi_m \to \mathcal{T}}$ based on the PTM-specific feature $\Phi_\mathcal{T}^m$ and target labels $\{y_i\}_{i=1}^N$ [66, 113, 55, 114]. Precise estimation requires a large $N$, which means we need to collect enough examples to identify the most helpful PTMs from a model zoo.

While the previous forward-based transferability assessment methods reduce the time cost, selecting among $M$ PTMs in the model zoo multiplies the forward cost $M$ times, making the estimation of $\hat{\boldsymbol{t}}$ computationally expensive. Moreover, since forward passes for complicated PTMs take longer, selecting a PTM *efficiently*, especially given a large model zoo, is crucial.

# 4 MODEL SPIDER

In MODEL SPIDER, we propose to tokenize PTMs and tasks regardless of their complexity, allowing us to *efficiently* calculate their relatedness based on a certain similarity measure over their representation, which capture general properties and serve as a specification of a model or task, demonstrating which kinds of tasks a model performs well on or what kind of models a task requires. In this section, we first introduce the process of obtaining repr. by learning from a training set of tasks, and the ability to rank PTMs could be generalized to downstream tasks. We then describe the encoder, the similarity measure, and an efficient way to generate supervision during representation learning. Finally, we discuss how MODEL SPIDER can be flexible in incorporating forward pass results of top-ranked PTMs to further improve the representation's semantics and the ranking's quality.

## 4.1 Learning to Rank PTMs with Representation

In MODEL SPIDER, we learn the model repr. $\{\boldsymbol{\theta}_m\}_{m=1}^M$, task repr. $\boldsymbol{\mu}(\mathcal{T})$, and the similarity measure $\mathrm{sim}(\cdot, \cdot)$ in a supervised learning manner based on a separate training set $\mathcal{D}$. The training set $\mathcal{D}$ does not contain overlapping classes with the downstream task $\mathcal{T}$.

Specifically, we randomly sample training tasks $\{\mathcal{T}_i\}$ from $\mathcal{D}$. For a given training task $\mathcal{T}_i$, we assume that we can obtain the ground-truth ranking $\boldsymbol{t}_{\mathcal{T}_i} = \{\mathrm{t}_{\phi_m \to \mathcal{T}_i}\}_{m=1}^M$ over the $M$ PTMs, indicating the helpfulness of each PTM. We will discuss the details of obtaining the supervision $\boldsymbol{t}_{\mathcal{T}_i}$ later. We then select PTMs for $\mathcal{T}_i$ based on the similarity between the task repr. $\boldsymbol{\mu}(\mathcal{T}_i)$ and those $M$ PTM repr. $\{\boldsymbol{\theta}_m\}_{m=1}^M$. We expect the higher the similarity, the more helpful a PTM is for the given task. We use $\Theta$ to denote all learnable parameters and optimize $\Theta$ with a ranking loss, which minimizes the discrepancy between the rank $\hat{\boldsymbol{t}}_{\mathcal{T}_i}$ predicted by the similarity function and the ground-truth $\boldsymbol{t}_{\mathcal{T}_i}$:

$$\min_\Theta \sum_{\mathcal{T}_i \sim \mathcal{D}} \ell_{\mathrm{rank}} \left( \hat{\boldsymbol{t}}_{\mathcal{T}_i} = \{\mathrm{sim}(\boldsymbol{\theta}_m, \boldsymbol{\mu}(\mathcal{T}_i))\}_{m=1}^M, \boldsymbol{t}_{\mathcal{T}_i} \right) . \tag{3}$$

Given $\boldsymbol{t} \in \mathbb{R}^M$, we use an operator $\mathrm{dsc}(\cdot)$ to index the elements of $\boldsymbol{t}$ in a descending order, *i.e.*, $\forall m < l$, we have $\boldsymbol{t}_{\mathrm{dsc}(m)} \geqslant \boldsymbol{t}_{\mathrm{dsc}(l)}$. $\mathrm{dsc}(m)$ is exactly the index of the PTM with $m$th largest

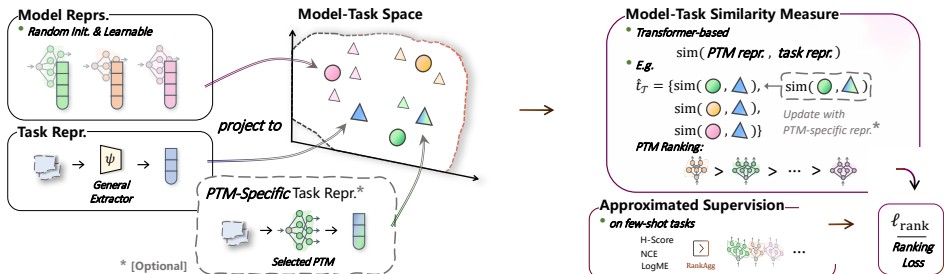

Figure 2: An illustration of MODEL SPIDER. The middle part (b) shows the workflow of MODEL SPIDER, which involves tokenizing both PTMs and tasks into a shared space. Plot (c) demonstrates how the model-task similarity calculated based on the representation helps rank PTMs for a given task. In plot (a), when the budget allows, MODEL SPIDER can take advantage of PTM-specific features obtained by performing forward passes of the top-$k$ ranked PTMs on some selected tasks. This improves the quality of task repr. as well as the PTM ranking.

ground-truth score. Based on this, we use the following ranking loss:

$$\ell_{\mathrm{rank}}(\hat{\boldsymbol{t}}, \boldsymbol{t}) = \sum_{m=1}^{M} - \log \left( \frac{\exp\left(\hat{\boldsymbol{t}}_{\mathrm{dsc}(m)}\right)}{\sum_{l=m}^{M} \exp\left(\hat{\boldsymbol{t}}_{\mathrm{dsc}(l)}\right)} \right) \ . \tag{4}$$

Equation 4 aims to make the *whole order* of the predicted $\hat{\boldsymbol{t}}_{\mathcal{T}_i}$ similar to the ground-truth $\boldsymbol{t}_{\mathcal{T}_i}$. So the similarity between the task repr. and that of a higher-ranked PTM indicated by $\boldsymbol{t}_{\mathcal{T}_i}$ should be larger than the similarity with lower-ranked PTM representation. The underlying intuition is that if a PTM performs well on certain tasks, it is likely to generalize its ability to related tasks. For example, if a PTM excels at bird recognition, it may effectively recognize other flying animals.

For a downstream task $\mathcal{T}$, we generate its task repr. with $\boldsymbol{\mu}(\mathcal{T})$, and identify the close PTM ones with the learned $\mathrm{sim}(\cdot, \cdot)$. Objective Equation 3 also works when the number of examples in a task is small. By learning to rank PTMs for sampled *few-shot tasks*, MODEL SPIDER can rank helpful models even with limited training data. We will show this ability of MODEL SPIDER in section 5.

## 4.2 Model and Task Representation for PTM Selection

We encode the general characteristics of tasks and PTMs via two types of representation.

**Model Representation.** Given a model zoo with $M$ PTMs, we associate a PTM $f_m$ with a form $\boldsymbol{\theta}_m \in \mathbb{R}^d$ encoding rich semantics about the aspects in which $f_m$ excels. Models pre-trained from related datasets or those with similar functionalities are expected to have similar representation.

**Task Representation.** A $C_{\mathcal{T}}$-class task $\mathcal{T} = \{(\boldsymbol{x}_i, y_i)\}_{i=1}^{N}$ contains a set of instances and labels. We would like to tokenize a task with a mapping $\boldsymbol{\mu}(\cdot)$, which outputs a set of vectors $\boldsymbol{\mu}(\mathcal{T}) \in \mathbb{R}^{d \times C_{\mathcal{T}}}$, one for each class. We implement $\boldsymbol{\mu}$ with one additional *frozen* encoder $\psi$ with an equivalent parameter magnitude as the PTMs in the model zoo. $\psi$ is pre-trained by self-supervised learning methods [17, 33, 53] and captures the semantics of a broad range of classes. In detail, we extract the features of all instances in the task $\mathcal{T}$ and take the class centers as the task repr.:

$$\boldsymbol{\mu}(\mathcal{T}) = \left\{ \frac{1}{|\mathbb{I}(y_i = c)|} \sum_{(\boldsymbol{x}_i, y_i) \in \mathcal{T}} [\psi(\boldsymbol{x}_i) \cdot \mathbb{I}(y_i = c)] \right\}_{c \in [C]} . \tag{5}$$

The task repr. expresses the characteristics of a task, *e.g.*, those tasks with semantically similar classes may have similar sets of representation.

**Model-Task Similarity.** The helpfulness of a PTM w.r.t. a task, *i.e.*, the transferability score, could be estimated based on the similarity of the model-task pairs $\hat{\mathrm{t}}_{\phi_m \to \mathcal{T}} = \mathrm{sim}(\boldsymbol{\theta}_m, \boldsymbol{\mu}(\mathcal{T}))$, and the PTM selection is complemented by embedding the model and tasks into a space and then identifying close PTM repr. for a task. In MODEL SPIDER, the $\mathrm{sim}(\cdot, \cdot)$ is implemented with a one-layer Transformer [98], a self-attention module that enables various inputs. The Transformer consists of alternating layers of multi-head self-attention, multi-layer perceptron, and layer norm blocks. We set the input of the Transformer as the union set of model and task repr. $\boldsymbol{z} = [\boldsymbol{\theta}_m, \boldsymbol{\mu}(\mathcal{T})] \in \mathbb{R}^{d \times (1+C)}$,

then the similarity $\hat{t}_{\phi_m \to \mathcal{T}}$ between model and task ones is:

$$\text{sim}(\boldsymbol{\theta}_m, \boldsymbol{\mu}(\mathcal{T})) = \text{FC}(\text{transformer}(\boldsymbol{z})[0]) \ , \tag{6}$$

where $[0]$ is the first output of the Transformer, *i.e.*, the corresponding output of the model representation. We add a Fully Connected (FC) layer to project the intermediate result to a scalar. Learnable parameters $\Theta$, including $\{\boldsymbol{\theta}_m\}_{m=1}^{M}$, FC, and weights of the Transformer, are trained via objective in Equation 3.

## 4.3 Accelerating Training for MODEL SPIDER

The training of MODEL SPIDER in Equation 3 requires a large number of (task $\mathcal{T}_i$, PTM ranking $\boldsymbol{t}_{\mathcal{T}_i}$) pairs. Although we could collect enough data for each task, obtaining the ground-truth PTMs rankings, *i.e.*, the helpfulness order of PTMs for each task, is computationally expensive. In addition, using some proxies of $\boldsymbol{t}_{\mathcal{T}_i}$ may weaken the ability of the MODEL SPIDER. We propose a closer approximation of the ground-truth $\boldsymbol{t}_{\mathcal{T}_i}$, which efficiently supervises sampled tasks from $\mathcal{D}$.

**Approximated Training Supervision**. We take advantage of the fact that existing PTM selection methods rely on the PTM-specific features $\Phi_{\mathcal{T}_i}^m$ to estimate the transferability score w.r.t. $\mathcal{T}_i$ and produce diverse scores. In other words, a PTM will be placed in different positions based on the scores provided by various methods such as NCE [97], LEEP [66], and LogME [113, 114]. Based on their "relatively good but diverse" ranking results, an intuitive approach to estimate the ground-truth $\boldsymbol{t}_{\mathcal{T}_i}$ is to *ensemble* their multiple ranking results into a stronger single order.

Given $\{\hat{\boldsymbol{t}}_{\mathcal{T}_i}^1, \hat{\boldsymbol{t}}_{\mathcal{T}_i}^2, \ldots\}$ as multiple predicted rankings over $M$ PTMs for a sampled task $\mathcal{T}_i$, *i.e.*, the order sorted by the estimations of transferability via various methods, we take advantage of Copeland's aggregation method [7, 82] to ensemble the orders: $\bar{\boldsymbol{t}}_{\mathcal{T}_i} = \{\bar{t}_{\phi_m \to \mathcal{T}_i}\}_{m=1}^{M} = \text{RankAgg}(\{\hat{\boldsymbol{t}}_{\mathcal{T}_i}^1, \hat{\boldsymbol{t}}_{\mathcal{T}_i}^2, \ldots\})$. Copeland's aggregation compares each pair of ranking candidates and considers all preferences to determine which of the two is more preferred. The output $\bar{\boldsymbol{t}}_{\mathcal{T}_i}$ acts as a good estimation of the ground-truth supervision $\boldsymbol{t}_{\mathcal{T}_i}$. The aggregated $\bar{\boldsymbol{t}}_{\mathcal{T}_i}$ is more accurate than a particular transferability assessment method, which improves the quality of the supervision in ranking loss in Equation 4.

**Sampling Tasks for Training**. We assume that the training data $\mathcal{D}$ contains a large number of classes with sufficient data. To sample tasks for training, we randomly select a set of classes from $\mathcal{D}$ and choose a subset of their corresponding examples. Benefiting from the supervision estimation approach $\text{RankAgg}$, we are able to obtain the aggregated ranking $\bar{\boldsymbol{t}}$ for any sampled task.

**Training Complexity**. The training phase in MODEL SPIDER is efficient. First, we pre-extract features $\{\Phi_{\mathcal{D}}^m\}_{m=1}^{M}$ for $\mathcal{D}$ with all PTMs in advance. Then only the computational burden of base transferability assessment methods, rank aggregation methods, and the optimization of top-layer parameters are involved. Furthermore, training tasks with the same set of classes share the same $\bar{\boldsymbol{t}}_{\mathcal{T}_i}$.

## 4.4 Re-ranking with Efficiency-Accuracy Trade-off

The learnable model representation captures the PTM's empirical performance on various fields of training tasks, which decouples the task repr. from the PTM. Each model repr. implicitly expresses the field in which the PTM excels, so the PTM selection only requires a task repr. to express the field in which the task is. In contrast to the general task repr. $\boldsymbol{\mu}(\mathcal{T}_i)$, PTM-specific features $\Phi_{\mathcal{T}_i}^m$ for a subset of PTMs provide *rich clues* about how those PTMs fit the target examples, which are also used in related transferability assessment approaches [25, 71]. We claim that given specific features with *a subset of PTMs* when the budget is available, our MODEL SPIDER can re-rank the estimated PTM order and further improve performance.

Specifically, we extract the PTM-specific task repr. $\boldsymbol{\mu}_m(\mathcal{T}) \in \mathbb{R}^{d_m \times C_{\mathcal{T}}}$ with the specific features $\Phi_{\mathcal{T}}^m$ of the $m$th PTM as Equation 5. To take account of different values of $d_m$ due to the heterogeneity of PTMs, we learn a projection $\mathbf{P} \in \mathbb{R}^{d_m \times d}$ for the $m$th PTM to align the dimensionality of $\boldsymbol{\mu}_m(\mathcal{T})$ with the model representation. We then replace the general task repr. $\boldsymbol{\mu}(\mathcal{T})$ via the specific one $\mathbf{P}_m^\top \boldsymbol{\mu}_m(\mathcal{T})$ when calculating the similarity with the repr. $\boldsymbol{\theta}_m$ of the $m$th PTM. The specific task repr. may facilitate obtaining more accurate estimations. During the training process, we dynamically select a partial set of PTMs and incorporate the specific repr. into the sampled tasks. Thus, the same Transformer module in Equation 6 can deal with the new type of representation. To differentiate the general and specific representation, we learn two additional $d$-dimensional embeddings as prompts.

Table 2: Performance comparisons of 10 baseline approaches and MODEL SPIDER on a model zoo with 10 PTMs [113]. We measure the performance with Kendall's [45] weighted $\tau_w$. The downstream tasks from diverse fields (8 datasets) are evaluated in a standard manner (all training examples) and a few-shot manner (10 examples per class and 30 trials). Specific features of top-3 ranked PTMs are used in MODEL SPIDER. We denote the best-performing results in bold.

| Method | Downstream Target Dataset | | | | | | | | Mean |
| | Aircraft | Caltech101 | Cars | CIFAR10 | CIFAR100 | DTD | Pets | SUN397 | |
|---|---|---|---|---|---|---|---|---|---|
| **Standard Evaluation** | | | | | | | | | |
| H-Score [9] | 0.328 | 0.738 | 0.616 | 0.797 | 0.784 | 0.395 | 0.610 | 0.918 | 0.648 |
| NCE [97] | 0.501 | 0.752 | 0.771 | 0.694 | 0.617 | 0.403 | 0.696 | 0.892 | 0.666 |
| LEEP [66] | 0.244 | 0.014 | 0.704 | 0.601 | 0.620 | -0.111 | 0.680 | 0.509 | 0.408 |
| $\mathcal{N}$-LEEP [55] | -0.725 | 0.599 | 0.622 | 0.768 | 0.776 | 0.074 | 0.787 | 0.730 | 0.454 |
| LogME [113] | **0.540** | 0.666 | 0.677 | 0.802 | 0.798 | 0.429 | 0.628 | 0.870 | 0.676 |
| PACTran [27] | 0.031 | 0.200 | 0.665 | 0.717 | 0.620 | -0.236 | 0.616 | 0.565 | 0.397 |
| OTCE [93] | -0.241 | -0.011 | -0.157 | 0.569 | 0.573 | -0.165 | 0.402 | 0.218 | 0.149 |
| LFC [25] | 0.279 | -0.165 | 0.243 | 0.346 | 0.418 | -0.722 | 0.215 | -0.344 | 0.034 |
| GBC [71] | -0.744 | -0.055 | -0.265 | 0.758 | 0.544 | -0.102 | 0.163 | 0.457 | 0.095 |
| MODEL SPIDER | 0.506 | **0.761** | **0.785** | **0.909** | **1.000** | **0.695** | **0.788** | **0.954** | **0.800** |
| **Few-Shot Evaluation (10-example per class)** | | | | | | | | | |
| H-Score [9] | -0.014 | 0.078 | 0.375 | 0.018 | 0.005 | -0.028 | -0.006 | 0.853 | 0.160 |
| NCE [97] | 0.273 | 0.534 | 0.597 | 0.267 | 0.232 | 0.362 | 0.352 | 0.793 | 0.426 |
| LEEP [66] | 0.069 | -0.038 | 0.476 | 0.530 | 0.471 | -0.111 | 0.567 | 0.468 | 0.304 |
| $\mathcal{N}$-LEEP [55] | -0.559 | 0.476 | 0.743 | 0.515 | 0.707 | 0.027 | 0.713 | 0.812 | 0.429 |
| LogME [113] | 0.341 | 0.453 | 0.497 | 0.718 | 0.698 | 0.407 | 0.657 | 0.817 | 0.574 |
| PACTran [27] | 0.136 | 0.262 | 0.484 | 0.631 | 0.614 | -0.227 | 0.701 | 0.477 | 0.385 |
| OTCE [93] | -0.316 | -0.050 | -0.127 | 0.515 | 0.505 | -0.168 | 0.406 | 0.210 | 0.123 |
| LFC [25] | 0.226 | -0.226 | -0.235 | 0.330 | 0.271 | -0.669 | -0.059 | -0.151 | -0.064 |
| MODEL SPIDER | **0.382** | **0.711** | **0.727** | **0.870** | **0.977** | **0.686** | **0.717** | **0.933** | **0.750** |

The prompts are added to the input repr., allowing the transformer to utilize represented-type context for a better ranking process. Notably, $\boldsymbol{\mu}_m\left(\mathcal{T}\right)$ depends on $\Phi_{\mathcal{T}}^m$, and the pre-extracted PTM-specific features for all training tasks make the construction of these specific representation efficient.

### 4.5 A Brief Summary of MODEL SPIDER

MODEL SPIDER learns to rank PTMs based on the model-task pair, balancing efficiency and accuracy. During the training, we sample tasks where PTM representation and transformer-based similarity are learned. In particular, to enable the model-task similarity to incorporate PTM-specific features, we replace some of the inputs to the transformer with enriched representations. We pre-extract PTM-specific features for all training tasks, and then the estimated ground-truth and the specific repr. could be constructed efficiently. During deployment, we first employ a coarse-grained PTM search with a general representation. Then we carry out forward passes over the target task *only for top-k ranked PTMs*, where the obtained PTM-specific task repr. will re-rank the PTMs by taking the distributed examples with PTM's features into account.

## 5 Experiments

We evaluate MODEL SPIDER on three benchmarks: the PTM zoo comprising heterogeneous models from the single-source, multi-source datasets, or composed of large language models. We analyze the influence of key components in MODEL SPIDER and visualize the ability of a PTM using spider charts based on the learned representation.

Table 1: Performance comparison of regression-conducted approaches with the same model zoo and weighted $\tau_w$ measurement as in Table 2. The downstream task is dSprites and UTKFace.

| Dataset | Methods for Regression Tasks | | | |
| | H-Score | LogME | GBC | Ours |
|---|---|---|---|---|
| dSprites | 0.106 | 0.612 | -0.283 | **0.679** |
| UTKFace | 0.075 | -0.156 | 0.052 | **0.364** |

### 5.1 Evaluation on a *Single-Source* Model Zoo

**Setups.** We follow [113] and construct a model zoo with 10 PTMs pre-trained on ImageNet [81] across five architecture families, *i.e.* Inception [88], ResNet [35], DenseNet [38], MobileNet [83],

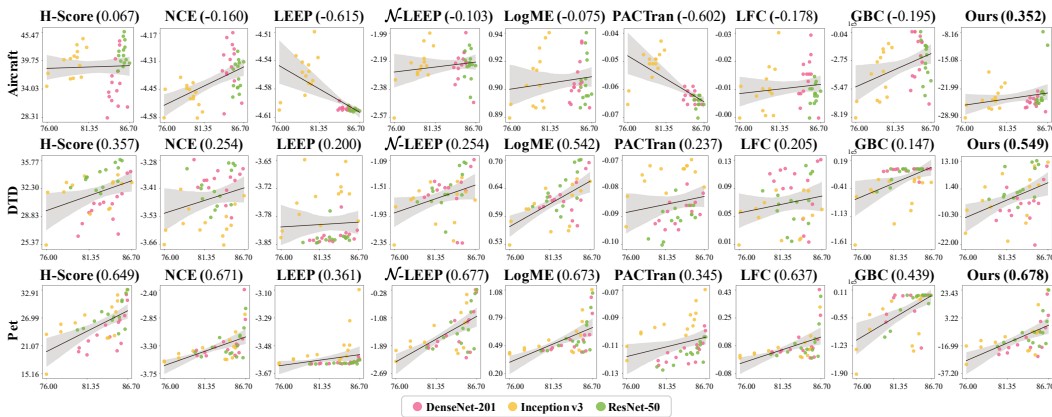

Figure 3: Visualizations when selecting PTMs from a multi-source heterogeneous model zoo (w/ 42 PTMs) on three downstream datasets. Rows represent approaches, and columns represent datasets. Correlations ($\tau_w$) are shown above each subfigure. The horizontal axis denotes transferred accuracy (w/ fine-tuning), while the vertical axis is the output ranking score. The PTM architectures are drawn in red, yellow, and green. The bold line and the gray area show the fitted straight line and the confidence interval for all PTMs. The strong linear correlation suggests superior performance.

and MNASNet [90]. We evaluate various methods on 9 downstream datasets, *i.e.* Aircraft [59], Caltech101 [32], Cars [47], CIFAR10 [49], CIFAR100 [49], DTD [19], Pet [73], and SUN397 [107] for classification, UTKFace [118] and dSprites [61] for regression.

**Baselines.** There are three groups of comparison methods. First are creating a proxy between PTM-specific features and downstream labels, such as H-Score [9], NCE [97], LEEP [66], $\mathcal{N}$-LEEP [55], LogME [113], and PACTran [27]. The second are based on the downstream inter-categories features like OTCE [93], Label-Feature Correlation (LFC) [25], and GBC [71]. Following [66] and [113], we equivalently modify NCE and H-Score to the general model selection application.

**Evaluations.** For the *standard evaluation*, we follow the official train-test split of each downstream dataset and utilize all the training samples. In *few-shot evaluation*, we consider if MODEL SPIDER can select useful models with limited labeled examples under privacy and resource constraints. We sample 10 examples per class from the training set as a "probe set" and report the average results over 30 trials. The full results, along with 95% confidence intervals, are presented in the appendix.

**Training Details of MODEL SPIDER.** We implement the $\psi$ with the pre-trained Swin-B [57, 53] to extract the task representation. MODEL SPIDER is trained on 832 sampled tasks from the mix of 6 datasets, *i.e.*, EuroSAT [36], OfficeHome [99], PACS [54], SmallNORB [51], STL10 [22] and VLCS [31]. MODEL SPIDER utilizes specific features from the top-3 ranked PTMs (out of 10) for downstream tasks, resulting in a 3-4 times speedup.

**Results of Standard and Few-Shot Evaluation.** For the standard evaluation shown in Table 2 and Table 1, MODEL SPIDER outperforms other baselines across datasets, except for Aircraft, which ranks top-2. It also demonstrates superior stability and outperforms all the existing approaches in few-shot scenarios, as displayed in the lower part of Table 2. Consistently ranking and selecting the correct PTMs, MODEL SPIDER achieves the highest mean performance among all methods.

## 5.2 Evaluation on a *Multi-Source* Model Zoo

We construct a large model zoo where 42 heterogeneous PTMs are pre-trained from multiple datasets.

**Setups.** PTMs with 3 similar magnitude architectures, *i.e.*, Inception V3, ResNet 50, and DenseNet 201, are pre-trained on 14 datasets, including animals [37, 46], general and 3D objects [32, 51, 49, 47, 14], plants [68], scene-based [107], remote sensing [106, 18, 36] and multi-domain recognition [54]. We evaluate the ability of PTM selection on Aircraft [59], DTD [19], and Pet [73] datasets.

**Training Details.** We use the same task representation extractor as in subsection 5.1 with 4352 training tasks sampled from the mix of the above datasets for pre-training the model zoo.

**Analysis of Multi-Source Model Zoo.** With many PTMs in the model zoo, we first set $k = 0$ and select PTMs based on general representation. We visualize the results in Figure 3, with each

Table 3: Top-1 ranked Large Language Model (LLM) performance comparisons against LLM evaluation results [94, 116, 96, 119, 69], which includes 2 directly baselines and our MODEL SPIDER, ranking on a pre-trained model zoo of 9 LLMs. The 10 downstream tasks are construct based on the OpenCompass [23] benchmark from 5 diverse fields as examination, language, knowledge, understanding, reasoning. We denote the best-performing results in bold.

| Method | Downstream Target Dataset | | | | | Mean |
|---|---|---|---|---|---|---|
| | Exam. | Language | Knowledge | Understand. | Reason. | |
| **LLM Evaluations** | | | | | | |
| Alpaca-7B [94] | 24.30 | 67.20 | 41.95 | 33.30 | 51.70 | 43.69 |
| ChatGLM2-6B [116] | 39.00 | 67.30 | 44.35 | 40.25 | 68.67 | 51.91 |
| LLaMA2-7B [96] | 31.30 | 67.40 | 55.90 | 40.30 | 52.93 | 49.57 |
| Vicuna-7B [119] | 29.10 | 66.70 | 49.45 | 34.70 | 52.67 | 46.52 |
| ChatGPT [69] | 39.90 | 60.90 | **57.10** | 55.40 | 69.90 | 56.64 |
| **Top-1 Results of LLM Ranking Methods,** *Selected by* | | | | | | |
| Self-assessed Confidence | 34.60 | 67.40 | 45.10 | 37.45 | 62.60 | 49.43 |
| Perf. on Similar Tasks | 29.10 | 67.20 | 44.35 | 53.45 | 63.03 | 51.43 |
| MODEL SPIDER | **41.30** | **67.65** | 55.90 | **56.80** | **70.07** | **58.34** |

subfigure showing the transferred accuracy using the selected PTM with fine-tuning and the predicted ranking score. A better-performing method will show a more obvious linear correlation. The results demonstrate that MODEL SPIDER achieves the optimum in all three datasets. Furthermore, a visualization of efficiency, the averaged performance over all datasets, and model size on this benchmark with standard evaluation is shown in Figure 1. The different configurations of $k$ balance the efficiency and performance in PTM selection, which "envelope" the results of other methods. These results confirm that MODEL SPIDER performs well in complex scenarios, highlighting its ability to select heterogeneous PTMs in a large model zoo.

## 5.3 Evaluation on a Zoo of *Large Language Models*

We introduce 9 open-source Large Language Models (LLMs) to construct our LLM zoo and deploy the MODEL SPIDER framework. We conduct a comparative analysis of the performance of the selected top-1 model against ChatGPT [69].

**Setups.** The LLM zoo involves Alpaca-7B [94], Baichuan-7B [109], Baichuan2-7B [109], ChatGLM2-6B [116], InternLM-7B [95], LLaMA2-7B [96], Vicuna-7B [119], Qwen-7B [8] and its chat fine-tuned version. We assess their zero-shot performance on diverse target tasks using the OpenCompass [23] LLM evaluation benchmark. We then focus on unseen tasks from the *examination* to *language*, *knowledge*, *understanding*, and *reasoning* datasets as the target tasks. For more details, please see appendix subsection B.3. We report the performance of the top-1 model recommended by each LLM ranking method and compare it with existing LLM evaluation results.

**Training Details.** For task representation, we employ a general Sentence-T5 [67] to obtain task representation. We extract answers from 10 instruction samples as a representative task for a dataset. We initialize the corresponding model repr. to encode the capabilities of LLMs on instruction data.

**Analysis of Ranking on the Zoo of LLMs.** Given that LLMs are computationally intensive in PTMs, we rank LLMs based on their general task representation. Intuitive methods for LLM ranking, like proxy measures relying on self-assessed confidence scores from generated answers or few-shot tasks in related domains, often fall short in assessing target task performance. The results indicate that while ChatGPT-3.5 demonstrates impressive performance in terms of universal performance across all diverse target tasks, as shown in Table 3 being 56.64, the top-1 ranked of MODEL SPIDER can surpass ChatGPT when efficiently choosing the appropriate LLM for each specific task. Our method achieves the average best and excels in the 4 out of 5 major fields of target tasks.

## 5.4 Ablation Studies

We analyze the properties of MODEL SPIDER on some downstream datasets, following the evaluation of a single-source model zoo in subsection 5.1.

**Will RankAgg provide more accurate ground-truth during training?** As discussed in subsection 4.3, MODEL SPIDER is trained on historical tasks and we utilize RankAgg to approximate

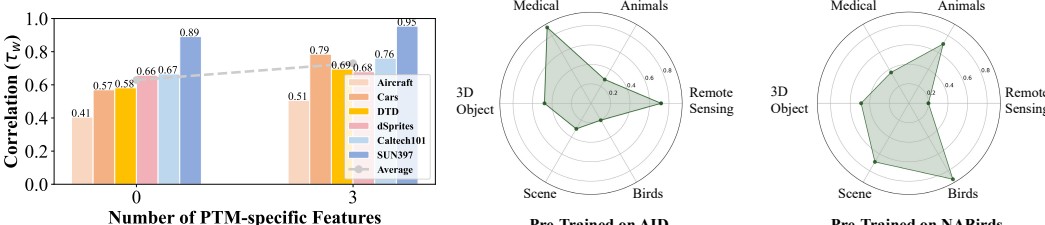

(a) Number of Used Features *vs*. Correlation.  (b) Spider chart of which semantic aspects the PTM excels in.

Figure 4: (a): The ablation analysis of how the ranking correlation changes (Y-axis) with more PTM-specific features (X-axis). (b): Visualization of the PTM's ability on 6 major semantic clusters of datasets with spider chart. The score on the vertex of the spider chart is the averaged similarities between a PTM and the task representation in the cluster. The higher the vertex value, the better a PTM would perform on that kind of task.

accuracy ranking. We investigate if this approximation offers better supervision and if using previous model selection methods like H-Score or LogME without aggregation is sufficient. The results in Table 4 include CIFAR10 and averaged results over eight classification datasets. It is evident that $\mathrm{RankAgg}$ provides stronger supervision during MODEL SPIDER's training.

**Will *more* PTM-specific features help?** As mentioned in subsection 4.4, MODEL SPIDER is able to incorporate PTM-specific features — the forward pass of a PTM over the downstream task – to improve the ranking scores. When no specific features ($k = 0$) exist, we use the general representation to rank PTMs (most efficient). In Figure 4 (a), we show that $\tau_w$ increases when MODEL SPIDER receives more PTM-specific features. It balances the efficiency and accuracy trade-off.

### 5.5 Interpreting MODEL SPIDER by Spider Chart

An interesting by-product of MODEL SPIDER is that we can visualize the ability of a PTM with a spider chart, which demonstrates which fields the PTM is good at. We cluster the datasets in our multi-source model zoo into six major groups. Then, we approximate a PTM's ability on the six types of tasks with the averaged similarity between a PTM to the tasks in the cluster. The larger the similarity, the better the PTM performs on that task. In Figure 4 (b), we find a PTM pre-trained on AID [106] dataset works well on medical and remote

Table 4: The weighted $\tau_w$ of MODEL SPI-DER variants when the training supervision is approximated by different methods. "Mean" denotes the averaged performance over 8 downsteam datasets in Table 2.

| Method | CIFAR10 | Mean |
|---|---|---|
| w/ H-Score [9] | 0.386 | 0.642 |
| w/ LogME [113] | 0.695 | 0.689 |
| w/ $\mathrm{RankAgg}$ (Ours) | **0.845** | **0.765** |

sensing tasks, and a PTM pre-trained on NABirds [37] dataset shows strong ability on birds and animal recognition. The spider charts provide valuable insights into PTM capabilities and assist in PTM recommendations for specific application scenarios.

## 6 Conclusion

The proposed MODEL SPIDER learns to rank PTMs for existing tasks and can generalize the model selection ability to unseen tasks, even with few-shot examples, and is applicable to both visual and large language models (LLMs). The two-stage pipeline in MODEL SPIDER enables it to fit the resources adaptively. A task is matched with PTMs efficiently based on their task-agnostic representation if resource is limited. While there is a sufficient resource budget, limited forward passes are carried out over the candidates of top-ranked PTMs, which re-ranks candidates via incorporating the detailed fitness between the task and the selected PTMs. The learned representations help construct a spider chart for each task, illustrating its relevance with all PTMs. The representation for models and tasks acts as a kind of specification that matches the main design in Learnware [121, 122, 91, 34, 92].

**Acknowledgments.** This work is partially supported by the National Key R&D Program of China (2022ZD0114805), NSFC (62250069, 62376118, 62006112, 62206245), Young Elite Scientists Sponsorship Program of Jiangsu Association for Science and Technology 2021-020, Collaborative Innovation Center of Novel Software Technology and Industrialization.

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

# Supplementary Material

We provide details omitted in the main paper.

- Appendix A: Workflow of MODEL SPIDER, encompassing the construction of model-task repr., training, and testing, with the "how to" and "answer" format.
- Appendix B: Experimental setups and implementation details of MODEL SPIDER, especially the two types of pre-training model zoos utilized in the experimental section.
- Appendix C: Additional experimental results conducted along different dimensions of robustness analysis.
- Appendix D: Additional datasets descriptions and other details mentioned in the main text.
- Appendix E: Discussions and future exploration of MODEL SPIDER.

## A    Details and Discussions of MODEL SPIDER

In the *method* section of the main text, we elucidate the comprehensive workflow for training and testing the deployment of MODEL SPIDER. This process encompasses **three main steps**, including (1) the extraction of task repr., (2) the extraction of model repr., and (3) the construction of a training scheme that assesses the ranking of matching between model-task repr., thereby establishing the ground-truth rank of the model zoo for a given task. Once these three steps have been accomplished, the subsequent phase entails training the MODEL SPIDER by leveraging the extracted repr. in conjunction with the ranked ground-truth information.

In essence, the testing and deployment strategy employed by the MODEL SPIDER framework epitomizes a balance between flexibility and efficiency. By employing a fixed feature extractor $\psi$ to acquire repr. pertaining to downstream target tasks, the trained MODEL SPIDER undergoes **a singular inference pass**, generating an output quantifying the similarity between each model and the downstream task representation. It then accomplishes the task of ranking the PTMs.

In the forthcoming sections, we elaborate on the details in the form of "*how to do it*" questions. The training process of MODEL SPIDER is illustrated in Algorithm 1, while the sampling procedure for training tasks is elaborated in detail in subsection A.2. Additionally, in subsection A.6, we expound upon the training strategy of PTM-Specific task representation. Analogously, the testing process of MODEL SPIDER is presented in Algorithm 2, and in subsection A.7, we provide a comprehensive exposition of the entire deployment workflow for ranking pre-trained models.

### A.1    How to construct model representations and task ones

This section supplements the details of subsection 4.2 and subsection 4.4, *i.e.*, the construction of the model-task repr., including the enriched PTM-specific ones.

**PTM representation.** The dimension of PTM repr., *i.e.*, the $d$ of $\boldsymbol{\theta} \in \mathbb{R}^d$ is implemented as $1024$. It is a learnable parameter that is optimized with the training process.

**Task representation.** The $\psi$ is implemented by a pre-trained Swin-B-based EsViT [57, 53] (linked at `https://github.com/microsoft/esvit`), self-supervised learning on the ImageNet-1K [81] with batch size $512$. In our experiments, this encoder acts as a wide-field feature extractor and is fixed without updating. The shape of task repr. $\boldsymbol{\mu}\left(\mathcal{T}\right) \in \mathbb{R}^{d \times C_{\mathcal{T}}}$ varies with the number of categories of downstream tasks. As mentioned in subsection 4.4, task reprs. enriched by the PTM-specific features are obtained through the forward pass of a PTM. We use another fully connected layer to project the PTM-specific feature to align with the model representation.

### A.2    How to sample the training tasks of MODEL SPIDER

We sample tasks for training MODEL SPIDER from additional datasets that are *disjoint* from the downstream tasks. These additional datasets possess notable differences and encompass diverse domains. Notably, MODEL SPIDER does not require substantial additional data for training. We sample the training tasks from a diverse pool of datasets. The number and size of the mixed datasets are controlled within a certain range. For more details, please see Appendix B.

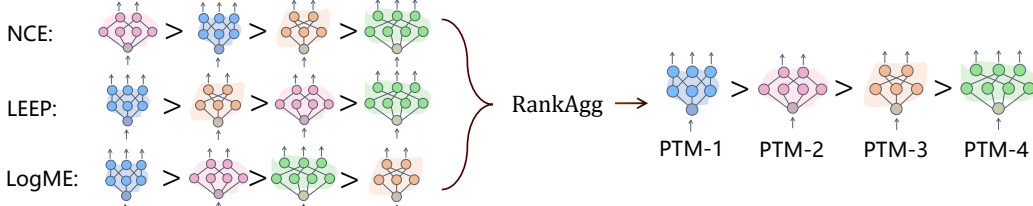

Figure 5: An illustration of the rank aggregation approach to ensemble the ranking of PTMs relying on diverse transferability assessment methods (three methods depicted in the figure). The PTMs that outperform more other PTMs should be placed ahead.

### A.3    How to see the relationship between RankAgg and MODEL SPIDER

We claim that RankAgg proposed by us cannot be considered as a direct baseline method. Firstly, RankAgg involves a substantial computational overhead when used as a stand-alone method for ranking PTMs. This is primarily due to the time and memory requirements of computing the base selection methods. Using RankAgg directly as a baseline would introduce a significant computational burden. However, we introduce RankAgg as an approximate ground-truth method for pre-computing in the training part of MODEL SPIDER. It is more efficient compared to full parameter fine-tuning.

Actually, MODEL SPIDER aims to demonstrate its broad generalization capacity by leveraging RankAgg to process an independent set of mixed data that has no overlap with the test data. This independent evaluation showcases the effectiveness of MODEL SPIDER in a real-world scenario and emphasizes its ability to handle diverse data efficiently. RankAgg itself does not play a role during the test execution of MODEL SPIDER.

### A.4    How to efficiently approximate the training ground-truth of MODEL SPIDER

This section complements subsection 4.4, wherein the training and ranking of the model zoo across multiple datasets are discussed. However, obtaining the ranking for all historical tasks through brute force is computationally expensive. To mitigate this issue, we introduce a rank aggregation method denoted as RankAgg, which serves as an approximation of the ground truth ranking.

Existing PTM selection methods rely on the PTM-specific features $\Phi_{\mathcal{T}}^m$ to estimate the transferability score. Different methods may have diverse score values — a PTM will be placed in different positions based on the scores provided by various methods. We empirically observe that some popular approaches such as NCE [97], LEEP [66], and LogME [113, 114] show "good but diverse" PTM ranking orders, so an intuitive approach to improving the transferability estimation quality is to *ensemble* their ranking results to a stronger single order.

As mentioned in subsection 4.3, given $\{\hat{\boldsymbol{t}}_1, \hat{\boldsymbol{t}}_2, \ldots, \hat{\boldsymbol{t}}_A\}$ as multiple rankings over the same set of $M$ PTMs for a target task $\mathcal{T}$, *i.e.*, the order sorted by the estimations of transferability via various methods, we take advantage of Copeland's aggregation method [7, 82] to ensemble the orders.

$$\bar{\boldsymbol{t}} = \{\bar{\mathrm{t}}_{\phi_m \to \mathcal{T}}\}_{m=1}^M = \mathrm{RankAgg}\left(\{\hat{\boldsymbol{t}}_1, \hat{\boldsymbol{t}}_2, \ldots, \hat{\boldsymbol{t}}_A\}\right) \ . \tag{7}$$

Copeland's aggregation compares each pair of ranking candidates and considers all preferences to determine which of the two is more preferred as illustrated in Figure 5.

Taking model $m, m'$ as an example, we define the *majority relation* to express the one-on-one dominance between these two models. Precisely, assuming that $A_m$ approaches rank model $m$ above model $m'$, *i.e.*, $\hat{\boldsymbol{t}}_{i,m} > \hat{\boldsymbol{t}}_{i,m'}$ with $A_m \times$ such $\hat{\boldsymbol{t}}_i$, while the remaining $A_{m'}$ ones do the opposite. Note that $A_m + A_{m'} = A$. The $m >_{\mathbb{M}} m'$ just in case $A_m > A_{m'}$, and correspondingly $m =_{\mathbb{M}} m'$ indicates $A_m = A_{m'}$. In summary, we define the aggregation score for model $m$ as:

$$\bar{\mathrm{t}}_{\phi_m \to \mathcal{T}} = \#\left\{i \mid m >_{\mathbb{M}} i\right\} + \frac{1}{2}\#\left\{i \mid m =_{\mathbb{M}} i\right\} \ , \tag{8}$$

where $\#\{\cdot\}$ is the size of the set. The aggregation score for a model is the number of others over which they have a majority preference plus half the number of models with which they have a preference tie. In our implementation, we aggregate the results of NCE, LEEP, LogME, and H-Score.

---

**Algorithm 1** The Training Part of the MODEL SPIDER

---

1: **Input:** fixed $\psi$, learnable parameters $\Theta$, including model repr. $\{\boldsymbol{\theta}_m\}_{m=1}^M$, FC for projection, and parameters of the transformer-based MODEL SPIDER

2: Sample training tasks $\{\mathcal{T}_i\}$ from the additional mixed datasets as in subsection A.2

3: Extract and save all task repr. $\bigcup_i \{\boldsymbol{\mu}(\mathcal{T}_i)\}$ with $\psi$.

4: **for all** sampled task $\mathcal{T}_i$ **do**

5:     **for** $m = 1$ **to** $M$ **do**

6:         **if** the $m$th PTM-specific features is available (randomly holds) **then**

7:             Derive the PTM-specific task repr. as mentioned in subsection 4.4.

8:
$$\hat{\mathrm{t}}_{\boldsymbol{\phi}_m \to \mathcal{T}_i} = \mathrm{sim}_\Theta\left(\boldsymbol{\theta}_m, \mathbf{P}_m^\top \boldsymbol{\mu}(\mathcal{T}_i)\right) .$$

9:         **else**

10:             Take model repr. $\boldsymbol{\theta}_m$ and estimate the similarity of model-representation pairs as Eq. 6.

11:
$$\hat{\mathrm{t}}_{\boldsymbol{\phi}_m \to \mathcal{T}_i} = \mathrm{sim}_\Theta(\boldsymbol{\theta}_m, \boldsymbol{\mu}(\mathcal{T}_i)) .$$

12:         **end if**

13:     **end for**

14:     *From above* `for`, the estimation scores of MODEL SPIDER $\hat{\boldsymbol{t}}$ is conducted.

15:     Calculate H-Score, NCE, LEEP, and LogME on $\mathcal{T}_i$.

16:     Aggregate on the results of existing approaches to obtain ground-truth $\bar{\boldsymbol{t}}$ as in subsection 4.3.

17:
$$\bar{\boldsymbol{t}} = \{\bar{t}_{\boldsymbol{\phi}_m \to \mathcal{T}_i}\}_{m=1}^M = \mathrm{RankAgg}(\{\hat{\boldsymbol{t}}_1, \hat{\boldsymbol{t}}_2, \ldots\}) .$$

18:     Optimize the parameters of MODEL SPIDER with ranking loss $\ell_{\mathrm{rank}}$ *w.r.t.* the ranking of $\bar{\boldsymbol{t}}$.

$$\ell_{\mathrm{rank}}(\hat{\boldsymbol{t}}, \boldsymbol{t}) = \sum_{m=1}^M -\log\left(\frac{\exp\left(\hat{\boldsymbol{t}}_{\mathrm{dsc}(m)}\right)}{\sum_{l=m}^M \exp\left(\hat{\boldsymbol{t}}_{\mathrm{dsc}(l)}\right)}\right) .$$

19:     Compute $\nabla_\Theta \ell_{\mathrm{rank}}$ and update corresponding parameters with the gradients

20: **end for**

21: **Output:** learned $\Theta$, including $\{\boldsymbol{\theta}_m\}_{m=1}^M$, FC, and parameters of the MODEL SPIDER

---

RankAgg can become quite time-consuming when calculating PTM ranking scores for the entire dataset, mainly due to the substantial overhead of computing the base selection methods. In our experimental setup, we integrate the RankAgg method as a module during the training phase, enabling us to pre-compute the rankings for each task. The RankAgg may raise the computational burden if employed directly as a testing baseline. Therefore, we employ RankAgg for the sampled *few-shot* tasks to balance ranking accuracy with efficiency and only use it in the training part. Note that MODEL SPIDER learns based on the RankAgg results, but is deployed independently of it and other baseline methods. Since RankAgg summarizes the PTM generalization capability on differentiated tasks spanning multiple domains, our model derived from the pre-aggregated rankings can learn the PTM ranking ability on a broader range of unseen tasks.

### A.5 How to learn the similarity of model-task representation

This section elaborates on subsection 4.1, *i.e.*, the learning process of MODEL SPIDER, especially the Transformer based estimation. **The Transformer-based module of model-task similarity.** The model-task repr. is concatenated as a sequence of features. The Transformer based module naturally fits and takes such input. Concretely, in operation, $\mathrm{transformer}(\cdot)$ is formalized as:

$$\mathrm{transformer}(\boldsymbol{z}) = \boldsymbol{z} + \boldsymbol{\alpha}(\mathtt{Q}, \mathtt{K}, \mathtt{V} = \boldsymbol{z})$$

$$= \boldsymbol{z} + \mathrm{softmax}\left(\frac{\boldsymbol{z}\mathrm{W}^{\mathtt{Q}} \cdot (\boldsymbol{z}\mathrm{W}^{\mathtt{K}})^\top}{\sqrt{d}}\right)\boldsymbol{z}\mathrm{W}^{\mathtt{V}} . \tag{9}$$

we apply linear projections on the query, key, and values using $\mathrm{W}^{\mathtt{Q}}$, $\mathrm{W}^{\mathtt{K}}$, and $\mathrm{W}^{\mathtt{V}}$, respectively. The similarity between prototypes is measured by the inner product in the transformed space, which

---

**Algorithm 2** The Downstream Inference Part of MODEL SPIDER

---

**Input:** target task $\mathcal{T}$, fixed $\psi$, learned $\Theta$
Obtain task repr. $\boldsymbol{\mu}(\mathcal{T})$ with $\psi$ as Eq. 5.
Estimate similarity of model-representation pairs as Eq. 6

$$\hat{\boldsymbol{t}} = \left\{ \hat{\mathrm{t}}_{\boldsymbol{\phi}_m \to \mathcal{T}} = \mathrm{sim}_\Theta(\boldsymbol{\theta}_m, \boldsymbol{\mu}(\mathcal{T})) \right\}_{m=1}^{M} .$$

Select top-$k$ PTMs via $\hat{\boldsymbol{t}} = \left\{ \hat{\mathrm{t}}_{\boldsymbol{\phi}_m \to \mathcal{T}} \right\}_{m=1}^{M}$.
Obtain the indexes in descending order via $\mathrm{dsc}(\cdot)$.
**for** $m = \mathrm{dsc}(1)$ **to** $\mathrm{dsc}(k)$ **do**
    Re-construct enriched repr. $\boldsymbol{\mu}_m(\mathcal{T})$, and update:

$$\hat{\mathrm{t}}_{\boldsymbol{\phi}_m \to \mathcal{T}} = \mathrm{sim}_\Theta\left( \boldsymbol{\theta}_m, \mathbf{P}_m^\top \boldsymbol{\mu}_m(\mathcal{T}) \right)$$

**end for**
**Output:** Rank PTMs with $\hat{\boldsymbol{t}} = \left\{ \hat{\mathrm{t}}_{\boldsymbol{\phi}_m \to \mathcal{T}} \right\}_{m=1}^{M}$

---

results in larger weights of the attention head $\boldsymbol{\alpha}$. Here $d$ is the size of every attention head. The output of the corresponding position of the model repr. is forwarding passed through a learnable MLP and then obtains the fitness estimated score of PTM selection.

**The learnable parameters in MODEL SPIDER.** To learn a PTM ranker, we optimize $M$ model repr. $\{\boldsymbol{\theta}_m\}_{m=1}^{M}$, the fully connected layer projection heads of the PTM-specific task repr. $\Phi_{\mathcal{T}_i}^m$ (mentioned in subsection 4.4) and the transformer-based model-task similarity evaluator $\mathrm{sim}(\cdot, \cdot)$, which is the main mapping and estimation module (mentioned in subsection 4.2).

### A.6 How to re-rank with PTM-specific task representation

As described in subsection 4.4 of the main text, we initially extract generic features using a fixed $\psi$ and conduct with the invariant task repr. across all PTMs. These features are used to generate a coarse-grained ranking by comparing the similarity between each task and the model representation. However, this ranking is solely based on a standardized task representation and does not account for the specific task-related information for each individual PTM.

Hence, we propose the re-ranking strategy specifically targeted at the top-k PTMs. During the testing phase, we leverage the coarse-grained ranking and perform inference on the downstream task with these top-k PTMs. Such PTM-specific task repr. are worked to update their similarity with the downstream task, as outlined in Algorithm 2. Notably, in the third line of the algorithm, we conduct a re-ranking based on the revised similarity scores obtained through this process.

### A.7 How to deploy MODEL SPIDER for testing

For a novel downstream task, we employ the generic feature extractor $\psi$ to extract the task representation. We then evaluate the similarity between each PTM in the model zoo and the given downstream task using the learned model repr. and a transformer-based MODEL SPIDER. If computational resources are available, we can leverage the results from the previous round to enhance the ranking process. Specifically, we can select the top-k PTMs from the previous ranking, extract their features, and apply the re-ranking approach as described in subsection A.6.

## B  Experimental Setups and Implementation Details

In this section, we introduce the experiment setups and implementation details, including constructing the pre-trained model zoo and training as well as deploying MODEL SPIDER.

## B.1 *Single-source* heterogeneous model zoo

**Construction of the model zoo.** We follow [113] and construct a model zoo with 10 PTMs pre-trained on ImageNet [81] across 5 families of architectures available from PyTorch. Concretely, they are Inception V1 [88], Inception V3 [88], ResNet 50 [35], ResNet 101 [35], ResNet 152 [35], DenseNet 121 [38], DenseNet 169 [38], DenseNet 201 [38], MobileNet V2 [83], and NASNet-A Mobile [90]. The model zoo spans PTMs of multiple parameter quantities. These pre-training models cover most of the supervised pre-training models the researchers employ.

**The downstream tasks.** There are 9 downstream tasks from various fields, including Aircraft [59], Caltech101 [32], Cars [47], CIFAR10 [49], CIFAR100 [49], DTD [19], Pets [73], and SUN397 [107] for classification, UTKFace [118] and dSprites [61] for regression. We use official train-test splits on each dataset and calculate the estimation scores for the baseline approaches on the training part.

**Transferred accuracy ranking of PTMs (ground-truth) after fine-tuning downstream tasks.** We follow You et al. [113] to obtain the ground-truth transferability score as well as the rankings $t = \{t_{\phi_m \to \mathcal{T}}\}_{m=1}^M$ ($M = 10$) with careful grid-search of hyper-parameters. Specifically, we grid search the learning rates (7 learning rates from $10^{-1}$ to $10^{-4}$, logarithmically spaced) and weight decays (7 weight decays from $10^{-6}$ to $10^{-3}$, logarithmically spaced) to select the best hyper-parameter on the validation set and compute the accuracy on the downstream test set. The training and computation of such a ground truth necessitates a substantial investment of over 1K GPU hours, imposing significant financial and computational burdens. Consequently, the feasibility of accomplishing this task within the constraints of training MODEL SPIDER is rendered unattainable.

**Sampling details of training tasks.** We sample the training tasks from a diverse pool of datasets. The datasets considered for sampling include EuroSAT, OfficeHome, PACS, SmallNORB, STL10, and VLCS. To ensure a representative training set, we randomly sample 832 tasks from all datasets. Each task is distributed across 2 to 4 mixed datasets and consists of 100 categories, and for each category, we randomly select 50 examples. In cases where the number of categories or examples to be sampled exceeds the specified limits, we select the maximum allowable value.

**Discussions.** This model zoo covers several classical structures commonly used in deep learning. The number of model parameters ranges widely, with large application potential. Still, there is also a situation where PTMs with larger scales tend to perform better in classification tasks and regression ones, making certain rankings always better on some datasets.

## B.2 *Multi-source* heterogeneous model zoo

**Construction of the Model Zoo.** As mentioned in the main text, we construct a large model zoo where 42 heterogeneous PTMs are pre-trained from multiple datasets in different domains, including animals [37, 46], general and 3D objects [32, 51, 49, 47, 14], plants [68], scene-based [107], remote sensing [106, 18, 36] and multi-domain recognition [54]. The concrete datasets are Caltech101 [32], Cars [47], CIFAR10 [49], CIFAR100 [49], SUN397 [107], Dogs [46], EuroSAT [36], Flowers [68], Food [14], NABirds [37], PACS [54], Resisc45 [18], SmallNORB [51] and SVHN [65]. The models' structures are 3 similar parameter-magnitude architectures, *i.e.*, Inception V3 [88], ResNet 50 [35] and DenseNet 201 [38]. The setting of the multi-source heterogeneous model zoo includes significantly more pre-training data than the single-source heterogeneous one described above. We pre-train the models with 3 structures on 14 datasets mentioned above ($3 \times 14 = 42$, initialized from the weights of the corresponding ImageNet pre-trained models).

**The downstream tasks.** We select 3 representative datasets as the downstream test tasks and conduct the PTM selection methods on them. Concretely, they are Aircraft [59], DTD [19] and Pets [73]. As outlined in the following description, we obtain the transferred fine-tuning accuracy (ground-truth) with an equivalent level of hyper-parameters search strategies.

**Transferred accuracy ranking (ground-truth).** Similarly, we adopt downstream supervised learning with optimizing by cross-entropy loss. We meticulously conduct a grid-search of hyper-parameters, such as optimizers, learning rates, and weight decays (2 optimizers as SGD or Adam, 6 learning rates from $5 \times 10^{-2}$ to $10^{-4}$, and 3 weight decay values from $5 \times 10^{-4}$ to $10^{-5}$, batch size of $128$, and the maximum epoch of 100). For the multi-domain dataset, like PACS [54], we set the test set to the same domain as the training set to reveal the in-domain performance. For the rest, we use the official train-test splits. We build the model zoo with around 5K GPU hours (on NVIDIA V100

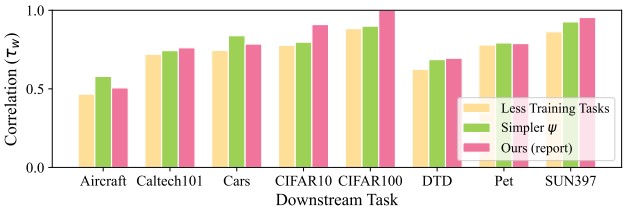

Figure 6: Ablation studies on simpler $\psi$ and less training tasks. We observed a slight decrease in performance when employing a weakened fixed feature extractor $\psi$ for MODEL SPIDER. Reducing the diversity of training tasks may result in performance degradation on some datasets.

GPUs). Similarly, when dealing with the expanded model zoo, the utilization of rigorous training methodologies to acquire the requisite ground truth for training MODEL SPIDER is eschewed.

**Sampling details of training tasks.** The sampling process for the multi-source heterogeneous model zoo is consistent with the single-source one mentioned above. In this case, we use the following datasets as the auxiliary set, *i.e.*, Caltech101, Cars, CIFAR10, CIFAR100, Dogs, EuroSAT, Flowers, Food, NABirds, PACS, Resisc45, SUN397, and SVHN. We randomly sample 4352 tasks for training.

**Discussion.** The availability of a multi-source heterogeneous model zoo introduces a wider array of models with varying structures, effectively covering a broader scope of domain knowledge. Consequently, this heightened diversity presents an increased difficulty in accurately ranking PTMs. Particularly, when a substantial gap exists between the characteristics of downstream tasks and the major PTMs, the ranking accuracy of some baseline methods undergoes a precipitous decline.

### B.3 *Large language models* zoo

**Construction of the model zoo.** We considered a setting for ranking pre-trained models in natural language processing, wherein we utilized a library of 9 commonly used open-source Large Language Models (LLMs). These LLMs include Alpaca-7B [94], Baichuan-7B [109], Baichuan2-7B [109], ChatGLM2-6B [116], InternLM-7B [95], LLaMA2-7B [96], Vicuna-7B [119], Qwen-7B [8] and its chat fine-tuned version. These open-source LLMs, trained by academic institutions or companies on vast corpora, possess robust zero-shot capabilities. However, compared to the performance on general tasks, LLMs have a domain gap regarding some specific tasks. Some new benchmarks [56, 40, 23] have been proposed recently to show that while ChatGPT [69] performs well concerning the average performance of general tasks, it may not consistently outperform other models in certain tasks. Additionally, the deployment complexity and computational costs of LLMs can vary significantly. Blindly choosing LLMs with large model sizes or high running expenses may not achieve optimal accuracy but is more likely to waste resources. Therefore, the urgent challenge is *accurately* and *efficiently* selecting the most suitable LLM for a given task within the available budget and constraints.

**The downstream tasks.** We focus on unseen tasks, *i.e.*, the *examination* datasets of AGIEval [120], as well as *language* datasets AFQMC [108], WSC [52], *knowledge* datasets BoolQ [20], NaturalQuestions [50], *understanding* datasets C3 [87], XSum [64], and *reasoning* datasets RTE [24], AX-b and AX-g [101] as the target tasks. When constructing tasks, whether for training or testing, we extract answers from 10 instruction data. During the generation of the *final* token in sequence generation, we extract features from the last layer. We calculate the average ranking score for 3 randomly sampled tasks from the target dataset as the final result.

**Sampling details of training tasks.** For training tasks, we deliberately choose different data from the test tasks, containing the remaining datasets in the OpenCompass [23] benchmark evaluation, such as GAOKAO-Bench [117], TyDiQA [21], CommonSenseQA [89], LAMBADA [72], COPA [79], and so on. We sample 10 instruction answers for each dataset, forming a task on that dataset. For each dataset, we sample a maximum of 16 training tasks, with mixed tasks from various datasets used in the training of MODEL SPIDER.

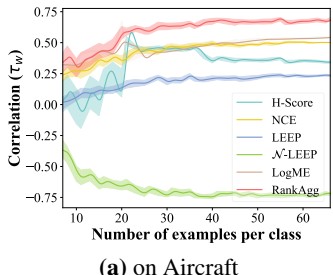
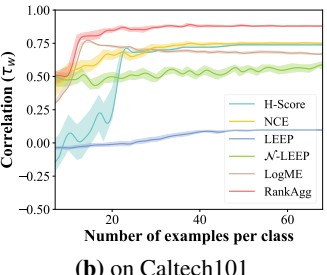

**(a)** on Aircraft  **(b)** on Caltech101

Figure 7: Correlation ($\tau_w$) given various number of examples per class on **(a)** Aircraft and **(b)** Caltech101. MODEL SPIDER shows stable and promising results in the low-shot scenario.

## C  Additional Experimental Results

### C.1  Ablation studies on simpler $\psi$ and less training tasks

We deploy additional experience with weakened conditions to verify the robustness of MODEL SPIDER. In Figure 6, we first introduce an attenuated simpler $\psi$, the additional encoder except for the PTMs in the model zoo. We import the tiny format pre-trained Swin-Transformer from EsViT (about this, please refer to subsection A.1 for more details). It has about half the number of parameters. The results show that although attenuated $\psi$ has only half of the parameters, it can still assist MODEL SPIDER in expressing task representation.

We then halve the training tasks to verify the significance of the training part diversity. We find that except for the performance degradation of the DTD dataset, the others are still flush with performance. MODEL SPIDER learns the characteristics of different PTM ability dimensions well despite the absence of training tasks.

### C.2  Ablation studies on the influence of training loss

As stated in the main text, the learning process of MODEL SPIDER incorporates a ranking loss. To assess the efficacy of this selection, alternative regression or ranking loss functions, such as mean square error (MSE) and ListMLE [105], are employed as replacements. The outcomes, presented in Table 5, clearly demonstrate that the presented ranking loss function surpasses the other alternatives in terms of both effectiveness and robustness. Notably, when alternative loss functions are utilized, the overall performance of MODEL SPIDER

Table 5: The weighted $\tau_w$ of MODEL SPIDER variants when the training objective is implemented by different loss functions. "Mean" denotes the averaged performance over 8 datasets.

| Method | CIFAR10 | Mean |
|---|---|---|
| w/ MSE | 0.558 | 0.526 |
| w/ ListMLE [105] | 0.777 | 0.735 |
| w/ $\ell_{\text{rank}}$ (Ours) | **0.845** | **0.765** |

experiences a substantial decline. These findings underscore the indispensable role of the ranking loss function within the framework of MODEL SPIDER.

### C.3  Ablation studies on the different shots of RankAgg and other baselines

We conduct an ablation analysis to compare RankAgg with several baseline methods on Aircraft and Caltech101 datasets with respect to the $\tau_w$ of the PTM ranking. We examined the variation of these metrics and their corresponding confidence intervals (in 95%) as the number of samples per class (shot) increased. The results, depicted in the provided Figure 7, are based on the average values and confidence intervals obtained from 30 randomly sampled sets for each shot. Due to computational constraints, certain baseline methods were omitted from the analysis. Notably, our findings reveal that the rank aggregation strategy effectively consolidates diverse perspectives on PTM ranking and consistently surpasses the performance of baselines across almost all shots.

Table 6: **Ablation studies** on the performance of MODEL SPIDER when the pre-trained model repository grows dynamically.

| MODEL SPIDER | Aircraft | Caltech101 | Cars | CIFAR10 | CIFAR100 | DTD | Pets | SUN397 | **Mean** |
|---|---|---|---|---|---|---|---|---|---|
| When the number of PTMs increases | | | | | | | | | |
| w/ number of 3 | 0.545 | 1.000 | 1.000 | 1.000 | 0.182 | 1.000 | 1.000 | 1.000 | 0.841 |
| increase to 6 | 0.573 | 0.627 | 0.818 | 0.905 | 0.839 | 0.445 | 0.888 | 0.336 | 0.679 |
| increase to 10 | 0.568 | 0.637 | 0.576 | 0.797 | 0.695 | 0.796 | 0.573 | 0.436 | 0.635 |

## C.4 Ablation studies on the dynamically incremental model zoo

When encountering new PTMs during the model selection task, the previously trained model repr. in MODEL SPIDER can be dynamically learned and updated. We employ an incremental learning approach [77] to address this challenge. Specifically, we sample 25% target tasks where the PTM ranking is closest to the average of all and insert the approximated accuracy of the new PTMs on them. This newly constructed ranking ground-truths include the correlation between old and new model repr., reducing the influence of imbalanced incremental data.

We performed ablation studies to investigate the behaviour of MODEL SPIDER as the pre-trained model zoo dynamically expanded. Our analysis focused on how can MODEL SPIDER could quickly adapt to newly added PTMs and integrate them into the ranking process. The results in Table 6 demonstrate that as the size of the model zoo increased from 3 to 6 and then to 10, MODEL SPIDER demonstrated the ability to incrementally learn the recommended ranking for the new additions to the model zoo. The incrementally learned ranking for the entire PTM zoo exhibited slightly lower accuracy than the results of direct training on all PTMs. Nonetheless, MODEL SPIDER consistently maintained an excellent level of performance.

## C.5 Confidence intervals for few-shot setting in Table 1 of the main text

We include the confidence intervals (in 95%) for the few-shot experiments in the respective section of Table 1 for the main text. These intervals were obtained through 30 repeated trials, providing a robust estimate of the performance variability in a few-shot manner.

## C.6 Illustration of re-ranking with PTM-specific task representation

In subsection 4.4, we discuss the learnable model repr., which captures the empirical performance of a PTM across various training tasks. This training scheme serves to decouple the task repr. from the forward pass of each PTM. Compared to the task repr. guided solely by general features, the PTM-specific task repr. provides more informative clues. By constructing it with the forwarding pass of PTM, we can incorporate the source PTM's adaptation information for downstream tasks. Our approach allows for the re-ranking of estimated PTM rankings using PTM-specific task representation. Since more forward passes consume more resources, MODEL SPIDER further improves performance and provides a dynamic resource adaptation option with PTM-specific features.

Illustrated in Figure 8 is an example of model re-ranking in the context of a heterogeneous multi-source model zoo. The MODEL SPIDER, after extracting PTM-specific task repr., accomplished a more precise PTM ranking. We re-construct the PTM-specific task repr. on the Dogs dataset pre-trained. Our investigation focuses on the Aircraft downstream dataset, and intriguingly, we discover that PTMs trained on multi-scenario multi-target datasets possessed inherent advantages when applied to the aircraft domain. This advantage can be attributed to their generally strong recognition capabilities for diverse targets. Remarkably, even models pre-trained on the Food dataset demonstrated exceptional performance on the Aircraft dataset. Despite the notable dissimilarities between the Food and Aircraft datasets, we conjecture that the Food-pre-trained models not only exhibit proficiency in recognizing multiple targets, encompassing various food items but also harbor latent potential for fine-grained recognition within the food domain. Consequently, these PTMs transfer their fine-grained recognition capacity to the aircraft domain. In contrast, the Dogs dataset, characterized by a narrow focus on a single biological species, impedes successful transfer to the Aircraft task.

Table 7: **The confidence interval (in 95%)** for few-shot evaluation (10 examples per class and 30 trials) in Table 1 of the main text. Specific features of Top-3 ranked PTMs are employed.

| Method | Downstream Target Dataset | | | | | | | |
|---|---|---|---|---|---|---|---|---|
| | Aircraft | Caltech101 | Cars | CIFAR10 | CIFAR100 | DTD | Pets | SUN397 |
| **Few-Shot Evaluation (10-example per class)** | | | | | | | | |
| H-Score [9] | $-0.014_{\pm 0.14}$ | $0.078_{\pm 0.13}$ | $0.375_{\pm 0.09}$ | $0.018_{\pm 0.12}$ | $0.005_{\pm 0.14}$ | $-0.028_{\pm 0.12}$ | $-0.006_{\pm 0.15}$ | $0.853_{\pm 0.02}$ |
| NCE [97] | $0.273_{\pm 0.05}$ | $0.534_{\pm 0.07}$ | $0.597_{\pm 0.02}$ | $0.267_{\pm 0.08}$ | $0.232_{\pm 0.04}$ | $0.362_{\pm 0.06}$ | $0.352_{\pm 0.09}$ | $0.793_{\pm 0.03}$ |
| LEEP [66] | $0.069_{\pm 0.04}$ | $-0.038_{\pm 0.01}$ | $0.476_{\pm 0.03}$ | $0.530_{\pm 0.04}$ | $0.471_{\pm 0.02}$ | $-0.111_{\pm 0.02}$ | $0.567_{\pm 0.02}$ | $0.468_{\pm 0.01}$ |
| $\mathcal{N}$-LEEP [55] | $-0.559_{\pm 0.06}$ | $0.476_{\pm 0.05}$ | $0.743_{\pm 0.04}$ | $0.515_{\pm 0.06}$ | $0.707_{\pm 0.03}$ | $0.027_{\pm 0.07}$ | $0.713_{\pm 0.04}$ | $0.812_{\pm 0.02}$ |
| LogME [113] | $0.341_{\pm 0.02}$ | $0.453_{\pm 0.01}$ | $0.497_{\pm 0.01}$ | $0.718_{\pm 0.02}$ | $0.698_{\pm 0.03}$ | $0.407_{\pm 0.01}$ | $0.657_{\pm 0.02}$ | $0.817_{\pm 0.00}$ |
| PACTran [27] | $0.136_{\pm 0.05}$ | $0.262_{\pm 0.02}$ | $0.484_{\pm 0.05}$ | $0.631_{\pm 0.02}$ | $0.614_{\pm 0.03}$ | $-0.227_{\pm 0.03}$ | $0.701_{\pm 0.03}$ | $0.477_{\pm 0.03}$ |
| OTCE [93] | $-0.316_{\pm 0.01}$ | $-0.050_{\pm 0.00}$ | $-0.127_{\pm 0.00}$ | $0.515_{\pm 0.00}$ | $0.505_{\pm 0.00}$ | $-0.168_{\pm 0.01}$ | $0.406_{\pm 0.00}$ | $0.210_{\pm 0.00}$ |
| LFC [25] | $0.226_{\pm 0.01}$ | $-0.226_{\pm 0.01}$ | $-0.235_{\pm 0.02}$ | $0.330_{\pm 0.04}$ | $0.271_{\pm 0.01}$ | $-0.669_{\pm 0.03}$ | $-0.059_{\pm 0.04}$ | $-0.151_{\pm 0.02}$ |
| Ours | $\mathbf{0.382}_{\pm 0.04}$ | $\mathbf{0.711}_{\pm 0.00}$ | $\mathbf{0.727}_{\pm 0.01}$ | $\mathbf{0.870}_{\pm 0.01}$ | $\mathbf{0.977}_{\pm 0.02}$ | $\mathbf{0.686}_{\pm 0.02}$ | $\mathbf{0.717}_{\pm 0.02}$ | $\mathbf{0.933}_{\pm 0.03}$ |

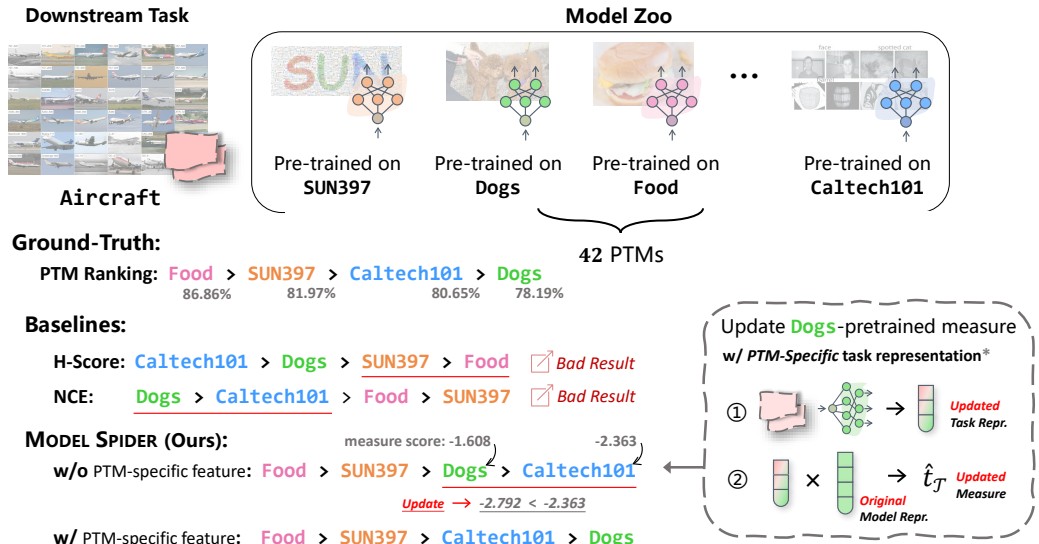

Figure 8: Illustrative re-ranking example with enhanced ranking through PTM-specific task representation.

The substantial disparities between the datasets pose a significant challenge for conventional baseline methods, often failing to prioritize the Food-pre-trained model. However, MODEL SPIDER successfully learns to rank the Food-pre-trained one and, through a meticulous screening process followed by result re-ranking, MODEL SPIDER identifies that the Caltech101-pre-trained model outperforms the Dogs-pre-trained one due to its superior multi-target recognition capabilities, thereby exhibiting enhanced transfer performance.

# D   More Details

## D.1   Comparison of the time consumption and memory footprint (details in Figure 1(c))

Figure 1(c) shows the average efficiency *vs* performance comparison over $5$ baseline approaches and MODEL SPIDER. The $k = 0$, $k = 3$, $k = 6$, $k = 36$, and $k = 42$ correspond to inference w/o PTM-specific features, w/ 3, 6, 36, and 42 ones. Following [113], we measure the wall-clock time (second) and memory footprint (MB) with code instrumentation.

## D.2   Datasets Description

We show the datasets description Table 9 with some examples Figure 9 covered in this paper.

Table 8: **Comparison of the time consumption and memory footprint** of fine-tuning, RankAgg, different baseline approaches, and MODEL SPIDER to rank the PTMs.

| Approaches | Wall-clock Time (*second*) | Memory Footprint (*MB*) |
|---|---|---|
| RankAgg | 7,318.06 | 10,405.32 |
| Fine-tuning (all parameters) | 614,497.22 | 13,872.81 |
| H-Score | 2,358.70 | 9,367.74 |
| NCE | 2,196.53 | 8,121.49 |
| LEEP | 2,215.06 | 8,209.33 |
| $\mathcal{N}$-LEEP | 4,963.01 | 9,850.84 |
| LogME | 2,571.99 | 8,217.80 |
| MODEL SPIDER (w/o PTM-Specific Feature) | 52.36 | 608.01 |
| MODEL SPIDER (w/ 3 PTM-Specific Feature) | 105.19 | 1,386.43 |
| MODEL SPIDER (w/ 6 PTM-Specific Feature) | 175.87 | 1,760.28 |
| MODEL SPIDER (w/ 36 PTM-Specific Feature) | 2,180.23 | 7,989.35 |
| MODEL SPIDER (w/ all (42) PTM-Specific Feature) | 2,402.77 | 9,954.09 |

Table 9: The number of training images, testing images and classes with the link to download the dataset.

| Dataset | Training Images | Testing Images | # Classes | URL |
|---|---|---|---|---|
| Aircraft [59] | 6,667 | 3,333 | 100 | https://www.robots.ox.ac.uk/~vgg/data/fgvc-aircraft/#aircraft |
| CIFAR10 [49] | 50,000 | 10,000 | 10 | https://www.cs.toronto.edu/~kriz/cifar.html |
| CIFAR100 [49] | 50,000 | 10,000 | 100 | https://www.cs.toronto.edu/~kriz/cifar.html |
| DTD [19] | 3,760 | 1,880 | 47 | https://www.robots.ox.ac.uk/~vgg/data/dtd/ |
| Stanford Cars [47] | 8,144 | 8,041 | 196 | https://ai.stanford.edu/~jkrause/cars/car_dataset.html |
| Caltech101 [32] | 3,060 | 6,084 | 101 | http://www.vision.caltech.edu/Image_Datasets/Caltech101/ |
| STL10 [22] | 5,000 | 8,000 | 10 | https://cs.stanford.edu/~acoates/stl10/ |
| Oxford Flowers 102 [68] | 2040 | 6149 | 102 | https://www.robots.ox.ac.uk/~vgg/data/flowers/102/ |
| CUB-200 [100] | 5994 | 5793 | 200 | http://www.vision.caltech.edu/visipedia/CUB-200-2011.html |
| Stanford Dogs [46] | 12,000 | 8,580 | 120 | http://vision.stanford.edu/aditya86/ImageNetDogs/ |
| EuroSAT [36] | 21,600 | 5,400 | 10 | https://github.com/phelber/eurosat |
| SmallNORB [51] | 24,300 | 24,300 | 5 | https://cs.nyu.edu/~ylclab/data/norb-v1.0-small/ |
| SVHN [65] | 73,257 | 26,032 | 10 | http://ufldl.stanford.edu/housenumbers/ |
| Food-101 [14] | 75,750 | 25,250 | 101 | https://www.tensorflow.org/datasets/catalog/food101 |
| NABirds [37] | 23,929 | 24,633 | 555 | https://dl.allaboutbirds.org/nabirds |
| NWPU-RESISC45 [18] | 25,200 | 6,300 | 45 | https://www.tensorflow.org/datasets/catalog/resisc45 |
| Oxford-IIIT Pets [73] | 3,680 | 3,669 | 37 | https://www.robots.ox.ac.uk/~vgg/data/pets/ |
| AID [106] | 8,000 | 2,000 | 30 | https://captain-whu.github.io/AID/ |
| PACS [54] | 5,446 | 616 | 7 | https://domaingeneralization.github.io/#data |
| VLCS [31] | 4,690 | 2,234 | 5 | https://github.com/belaalb/G2DM#download-vlcs |
| Office-Home [99] | 11,231 | 11,231 | 65 | https://www.hemanthdv.org/officeHomeDataset.html |
| SUN397 [107] | 87,003 | 21,751 | 397 | https://vision.princeton.edu/projects/2010/SUN/ |
| ImageNet-1K [81] | 1,281,167 | 50,000 | 1000 | http://image-net.org/download |

# E   Discussions

There are two promising directions of MODEL SPIDER. First, MODEL SPIDER exhibits the unique characteristic of not relying on the forward pass of the model zoo, thereby enabling the evaluation of task compatibility with *classical machine learning models*. Then, MODEL SPIDER could be applied to the case when we use *other criteria* in addition to fine-tuning performance to measure the fitness between a model and a task.

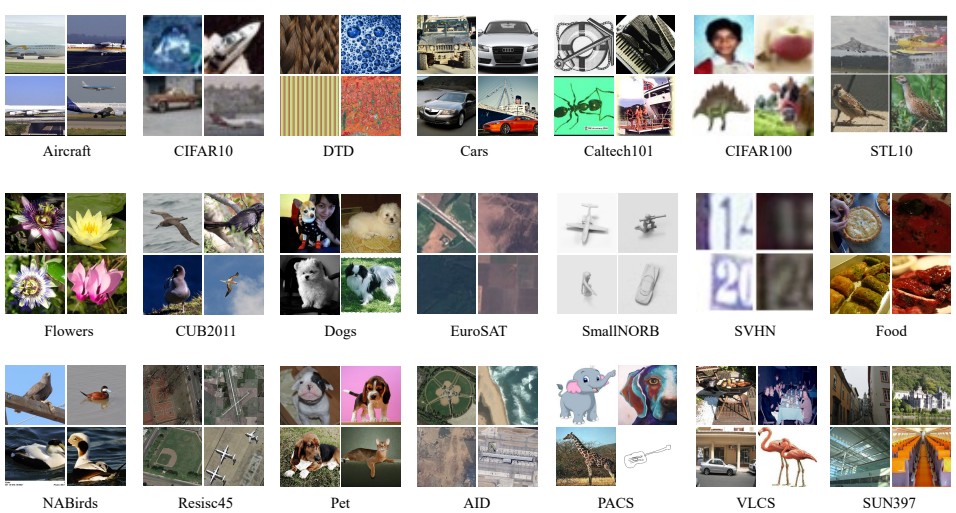

Figure 9: Examples of datasets.

