# OpenReview forum: "Model Spider: Learning to Rank Pre-Trained Models Efficiently"
_NeurIPS.cc/2023/Conference — NeurIPS 2023 spotlight_

### Official Review · Reviewer_mKeC · 2023-06-28

**Soundness:** 2 fair
**Presentation:** 3 good
**Contribution:** 2 fair
**Rating:** 5
**Confidence:** 4

**Summary:**

This paper investigates how to select the most suitable PTM given a target task efficiently and accurately.
A novel approach called Model Spider has been proposed. It learns to encode both PTMs and tasks into vectors and measures their similarity, which is further used to rank the PTMs. It can also incorporate task-specific forward results of PTMs for more accurate re-ranking when resources budgets allow.
Extensive experiments have been conducted to verify the effectiveness of the proposed method.

**Strengths:**

1. The idea of encoding PTMs and downsteam tasks into vectors for PTM ranking is well-motivated.
2. The overall presentation of the proposed method is well-organized and generally easy to follow.

**Weaknesses:**

1. The generalization ability of the proposed method has not been well-verified. The supervised training method is known to result in models with poor generalization ability. It seems that the proposed method heavily relies on the frozen encoder $\psi$ to capture the relevance between different tasks. What if the new task is quite different from the tasks used in training? When evaluating the proposed method, all the downstream tasks are about image classification. The authors should evaluate the generalization ability of the proposed method with more diverse downstream tasks.
2. The proposed method is highly related to the classification task as it uses the class centers as the task token in Equation (5). However,  many tasks do not have such "classes", such as regression tasks and generation tasks. How to adopt the proposed method to such tasks has not been elaborated.
3. Only 10 PTMs are used when evaluating the proposed method on a single-source model zoo, which is not enough. It is recommended to evaluate the proposed method in the NLP domains, where there are numerous pre-training models and diverse downstream tasks, which can be used to better verify the generability of the proposed method to a new task.


**Questions:**

In addition to the questions above, I have the following questions:
1. What's the motivation for using Equation (4) to train the model to capture the ranking order? There are many other methods that can train the model to learn to rank, such as the pairwise BPR and listwise ListMLE. It is recommended to verify the impact of different ranking loss functions.
2. Does the proposed method require re-training whenever a new PTM comes? If so, the proposed method seems costly when applied in real-world scenarios.

**Limitations:**

No, the authors have not discussed the limitations of the proposed method. The authors are recommended to explain whether the proposed method is limited to classification tasks and how to adapt it to other tasks.

---

> ### Author Rebuttal · Authors · 2023-08-10
>
> **Thank Reviewer mKeC for the valuable insights and thoughtful questions**. The feedback enhances our work's clarity and robustness. Here is our response:
>
> **Question 1**: Generalization ability and dependency on frozen encoder $\psi$. The new task differs from the training ones.
>
> **Answer 1**: **Our method, Model Spider, showcases broad task generalization for pre-trained model ranking. It minimizes dependency on the potency of $\psi$**. Model Spider learns task tokens from $\psi$, reflecting task importance relative to pre-trained models, **not an absolute need for exceptional task representation**. In Appendix Figure 2 and line 194, we introduce an attenuated version of $\psi$ in a tiny scale and conduct ablation studies(green vs. pink in the figure).
>
> Moreover, when the downstream task significantly differs from the tasks during training, Model Spider maintains robust performance.
>
> + **New Field Tasks in Natural Language Processing**
>
> Transitioning from visual recognition to NLP, **we employ rule-based task categorization to create NLP sub-zoos**, refining model ranking. We demonstrate the ranking capabilities in NLP and large language model contexts in Tables 2 and 3 of the general response. Model Spider continues to showcase exceptional applicability and generalization.
>
> + **New Domain Tasks with Out-of-Distribution**
>
> For downstream tasks beyond training distribution, the general feature extractor $\psi$ might have limited representation capacity for that particular task. To illustrate this, we conduct experiments that analyze $\psi$'s performance on certain datasets. We extract features using $\psi$ and compare the distances to training class centers, akin to Nearest Class Mean (NCM) [1]. In the following, we compare the accuracy (in %) differences between full-parameter fine-tuning and NCM on various datasets. Larger differences indicate more disparities in downstream tasks. Notably, datasets like DTD and Pet, **which deviate notably from the pre-training data distribution, highlight $\psi$'s limited representation capacity in such cases**.
>
> | Method | CIFAR-100 | Caltech-101 | DTD | Pet | EuroSAT |
> |-------|-------|-------|-------|-------|-------|
> | Fully Fine-tuning | 69.66 | 85.62 | 63.39 | 84.35 | 93.62 |
> | Feature-based NCM | 66.41 | 85.80 | 57.19 | 78.64 | 89.11 |
>
> As shown in the Table 1 of the paper, we observe that Model Spider still demonstrates exceptional performance on DTD and Pets datasets.
>
> Furthermore, **we extend our evaluation to the EuroSAT remote sensing dataset**, demonstrating Model Spider's effective generalization ability.
>
> | Method | EuroSAT |
> |-------|-------|
> | LEEP | 0.395 |
> | LogME | 0.510 |
> | Model Spider | **0.682** |
>
> + **New Type Tasks from Regression**
>
> As shown in Table 2 of the paper (line 277), Model Spider excels in regression tasks, capturing relative model proficiency order.
>
> The $\psi$ captures **the relative order of task representation rather than absolute task-specific information**. It's most effective when a semantic connection exists between models and tasks, enhancing model ranking. Furthermore, for the diagram spanning pre-training to fine-tuning, the pre-training knowledge is maximally stimulated when downstream tasks are semantically related to models. Otherwise, the fine-tuned performance of unrelated pre-trained models resembles that of randomly initialized models.
>
> &nbsp;
>
> **Question 2**: Task representation constrained by class centers.
>
> **Answer 2**: In our method, task tokens encapsulate task representations, with various implementation options. The class center is just one specific to classification. **Even without categorical information in the target tasks, meaningful task representation can be established by forming semantically relevant prototype clusters.** For instance, in regression tasks (e.g., dSprites, UTKFace in Table 2), Model Spider constructs semantic clusters representing age and position, yielding favorable results.
>
> We explore alternative task representation. In DTD dataset of Figure 3, we sample around 235 samples and compute the covariance of these samples. The results are as follows:
>
> | Method | DTD |
> |-------|-------|
> | Prototype Token | **0.549** |
> | Covariance Token | 0.513 |
>
> In conclusion, Model Spider is capable of learning from diverse task representations.
>
> &nbsp;
>
> **Question 3**: Generability on larger pre-trained model zoo or to new tasks.
>
> **Answer 3**: In Figure 3 of the paper, we extend the model zoo from Table 1 (consisting of 10 pre-trained models, following LogME) **to 42 pre-trained models. This extended zoo covers three similar magnitude architectures**: Inception V3, ResNet 50, and DenseNet 201, each pre-trained on 14 datasets (lines 288-292).
>
> Furthermore, NLP and large language models are evaluated (Tables 2 and 3 of the general response), see Answer 1 for more details. Model Spider consistently demonstrates exceptional performance.
>
> &nbsp;
>
> **Question 4**: The motivation for Equation (4). Comparison to the pairwise BPR and listwise ListMLE.
>
> **Answer 4**: Building on the cross entropy loss which aims to boost the position of the one-hot label, we craft a multi-round optimization where, in round $m$, we elevate the $m^{th}$ largest item above items from $m + 1$ to $M$. We pick the $m^{\text{th}}$ item using the $\mathrm{dsc}(\cdot)$ operator. Also, the denominator summation only includes items from $m$ to $M$ (not all). These nuances distinguish our ranking loss from ListMLE. **Our optimization moves from local to global ranking, capturing richer contextual ordering**. Unlike pairwise BPR which only compares items, our loss considers global ranking, reflecting the entire distribution. We include ListMLE results in Table 1 of the Appendix.
>
>
> **Thank Reviewer mKeC very much for the valuable suggestions. We will incorporate the relevant content into the final version**.
>
> [1] Distance-Based Image Classification: Generalizing to New Classes at Near-Zero Cost.

---

> ### Comment · Reviewer_mKeC · 2023-08-14
>
> I appreciate the authors' efforts during the rebuttal phase. I have carefully read the reviews from other reviewers and the authors' corresponding responses. I thank the authors for the detailed answers to my review, which have resolved my primary concerns. I'd like to raise my rating to borderline accept regarding the novelty of this work.

---

> > ### Author Response · Authors · 2023-08-14
> >
> > We are sincerely grateful for the thoughtful revisions by Reviewer mKeC. We will persist in our efforts moving forward. Thank you very much.

---

### Official Review · Reviewer_1u8a · 2023-07-04

**Soundness:** 3 good
**Presentation:** 3 good
**Contribution:** 3 good
**Rating:** 5
**Confidence:** 3

**Summary:**

This paper introduces Model Spider, a unique method to efficiently and accurately rank Pre-Trained Models (PTMs) for a specific task within a model zoo. Model Spider innovatively creates tokens for both PTMs and tasks, encapsulating their characteristics in a manner that facilitates an efficient selection process. It utilizes a separate set of training tasks to learn how to construct these tokens and calculate the fitness score between a model-task pair. The paper also presents a strategy to update the task tokens based on the semantics specific to the top-ranked PTM candidates, improving the final selection. The key contributions of this paper are the innovative method of tokenizing tasks and PTMs for easy ranking, and the ability of the system to incorporate task-specific forward results of certain PTMs within resource limitations. Through rigorous testing across various model zoo configurations, the authors demonstrate the efficacy of Model Spider, showing significant improvements in PTM selection and efficiency. The work represents an innovative solution to the challenge of sifting through the proliferating number of PTMs to find the most suitable model for a given task.

**Strengths:**

1. Proposed methods show quite significant improvements over the strong baselines.
2. Methodology of tokenizing PTM looks new. PTMs are tokenized by mapping them to unsupervised trained task tokens. The method looks sensible.
3. Applying learning to rank to rank the model fitness is also new and interesting.
4. Extensive experiments with ablation studies.

**Weaknesses:**

1. “Hyperparameter k” in Figure 1 is not explained at all. In the main text, notation k is also not clearly defined, only top-k mentioned here and there. Readers have to guess k means the number of top ranked Pretrained Models.
2. Some important results and discussions are included in the Appendix, but no reference in the main text. Please refer readers to the appendix for the useful information.
3. Figure 3 is a bit hard to read. What information should I draw from the figure, besides correlation number?

**Questions:**

1. How good can the method be applied to the pre-trained large language models?

**Limitations:**

1. Good tokenization of pre-trained models (PTMs) depends highly on the number of high-quality PTMs. How the Model spider perform when the number of PTMs M and the number of tasks varies?
2. The training of Model Spider depends on the RankAgg. As authors explained (in the appendix), RankAgg introduces a significant computational burden. How could one get a RankAgg model to be used for training Model Spider in the first place?

---

> ### Author Rebuttal · Authors · 2023-08-10
>
> **We deeply appreciate Reviewer 1u8a's insightful queries and constructive input**. The engagement has undoubtedly enhanced the quality and coherence of our paper. Here are our responses:
>
> **Question 1**: Hyperparameter $k$ in Figure 1.
>
> **Answer 1**: The hyperparameter $k$ corresponds to **the number of top-ranked pre-trained models (lines 51, 249, and 319)**. In Section 4.4 "Re-ranking with Efficiency-Accuracy Trade-off," we introduce a two-step process in Model Spider. Initially, a general task token is used for the initial ranking of pre-trained models. Then, the top-$k$ **pre-trained models perform forward passes on the target task**. This generates PTM-specific task tokens, aiming to enhance our ranking outcomes. For instance, with a smaller value of $k$, Model Spider exhibits high efficiency in model ranking, confirmed by the results in Appendix Table 4: "Comparison of the time consumption and memory footprint." Moreover, in lines 319-323 of the main text and Figure 4, we demonstrate that as $k$ increases, the overall ranking accuracy also improves.
>
> &nbsp;
>
> **Question 2**: Refer readers to the Appendix for useful information.
>
> **Answer 2**: Thank Reviewer 1u8a for the valuable reminder. We will include relevant references in the final version, such as comparing **the time consumption and memory footprint of various $k$, other related ablation studies**, and so on.
>
> &nbsp;
>
> **Question 3**: The information in Figure 3, besides the correlation number.
>
> **Answer 3**: In Figure 3, we can observe **the direct linear relationship between the predicted model ranking and the downstream fine-tuned accuracy**. The x-axis represents the fine-tuned accuracy of the corresponding pre-trained models, and the y-axis reflects the scores predicted by the model ranking method. Higher scores imply a higher chance of superior performance. For instance, the better result, like Ours with a score of 0.678, on the Pet dataset compared to LEEP with a score of 0.361 exhibits the fitting line that closely approximates a slope of $1$. **The values of the points along the y-axis offer insights into the distribution of the method's evaluated scores**. Different colors of points correspond to different model structures. **Multiple points of the same color stem from diverse tasks during pre-training**. For example, ResNet-50 models mostly perform well on the Aircraft dataset, whereas this trend is not necessarily consistent on the Pet dataset.
>
> &nbsp;
>
> **Question 4**: Application to the pre-trained large language models.
>
> **Answer 4**: We explore the performance of Model Spider in NLP tasks and large language models. **The results are provided in Tables 2 and 3 of the general response**. We sincerely appreciate the forward-looking suggestions from the reviewers, and **we will certainly incorporate the relevant experiments in the final version of the paper** to achieve even greater advancements in the future.
>
> &nbsp;
>
> **Question 5**: How the Model Spider perform when the number of PTMs $M$ and the number of tasks varies.
>
> **Answer 5**: In Table 2 of the Appendix, we conducted ablation studies on the performance of Model Spider as the size of the pre-trained model zoo dynamically increases. It is observed that changes in the number of models within the pre-trained model zoo can indeed influence the performance of Model Spider, with **greater model diversity posing increased challenges for the model ranking task**.
>
> In Table 1 of the general response, we further **expanded the model repository by including larger models like ViT-B/16**. This extended coverage enhances the diversity of the model zoo. Model Spider continues to demonstrate consistently high-performance levels in this enriched setting.
>
> Additionally, concerning the scope of training tasks, we downsized the training task set for training Model Spider. **The results of this reduction are presented in the Appendix at line 194 and Figure 2 (yellow compared to pink)**. Our results show that, apart from a slight drop in performance on the DTD dataset, **Model Spider maintains strong overall performance even with fewer training tasks**. This underscores Model Spider's ability to effectively capture diverse pre-trained model characteristics despite reduced training task diversity. These findings highlight the robust performance of Model Spider across different numbers of pre-trained models and tasks.
>
> &nbsp;
>
> **Question 6**: RankAgg introduces a significant computational burden. How could one get a RankAgg model to be used for training Model Spider?
>
> **Answer 6**: As mentioned in lines 308 of the main text and 48 of the Appendix, RankAgg requires pre-computation and entails some overhead. However, **it is significantly more efficient compared to full parameter fine-tuning.**
>
> In the experiment detailed in **line 69 of the Appendix and depicted in Appendix Figure 3**, we elaborate on the motivation behind introducing RankAgg. We empirically observe that popular approaches such as NCE, LEEP, and LogME exhibit "good but diverse" pre-trained model ranking orders. Consequently, ensembling their ranking outcomes into a stronger single order appears to be an intuitive way to enhance transferability estimation quality.
>
> The algorithmic process of RankAgg is **outlined in Algorithm 1 (line 16) of the Appendix, as well as in lines 73 and 83**. It involves initially sampling a subset of training tasks containing partial samples, calculating the model rankings derived from NCE, LEEP, LogME, and H-Score methods, and then **aggregating these rankings using RankAgg to obtain improved results**. The RankAgg method is designed to be plug-and-play, demonstrating excellent scalability and ease of use.
>
> **We sincerely thank the Reviewer 1u8a for the valuable insights**. We are fully committed to incorporating these elucidating descriptions into the final version of the paper.

---

> > ### Comment · Reviewer_1u8a · 2023-08-14
> >
> > Thank you for replying my questions. I would keep voting for borderline accept.

---

> > > ### Author Response · Authors · 2023-08-15
> > >
> > > We genuinely thank Reviewer 1u8a for the valuable support. We will continue to make revisions accordingly. Thank you very much.

---

### Official Review · Reviewer_iJup · 2023-07-05

**Soundness:** 3 good
**Presentation:** 3 good
**Contribution:** 3 good
**Rating:** 7
**Confidence:** 3

**Summary:**

This paper introduces a very interesting approach named "model spider", to address the challenging problem of selecting suitable Pre-Trained Models (PTMs) from a large number of options to fit the target tasks. Instead of relying on time-consuming and computationally heavy forward or backward passes over all PTMs, the model spider tokenizes both PTMs and tasks, summarizing their characteristics into vectors for efficient PTM selection. Experiments show that model spider performs well in various configurations of model zoos, providing a balance between efficiency and selection accuracy.

**Strengths:**

1. The problem and idea of this paper are quite interesting.
2. The proposed method is simple yet effective.
3. The results are quite encouraging.

**Weaknesses:**

1. The results are tested on vision tasks. It is not clear whether the proposed method can be generalized to tasks in other modalities.

2. The model size used in this work is relatively small (only up to tens of millions of parameters). It is not clear whether the proposed method can handle larger models, such as ViT, BERT, and GPT. It would be more exciting if larger models (especially large language models) can be easily evaluated.

3. The proposed method still requires some samples to train on new tasks. It would be better to consider using some methods (e.g., meta-learning) to generate initial tokens for tasks so that new tasks can be handled without additional training samples.

Minor:
The term "token" is a little confusing since the meaning of token in this paper is different from the common meaning of tokens in PTMs.

**Questions:**

See weaknesses.

**Limitations:**

The authors have discussed the limitations.

---

> ### Author Rebuttal · Authors · 2023-08-10
>
> **We sincerely appreciate Reviewer iJup's perceptive suggestions and valuable feedback**. The suggestions have been instrumental in enhancing our paper. For some questions, our responses are as follows:
>
> **Question 1**: Whether the proposed method can be generalized to tasks in other modalities.
>
> **Answer 1**: Thank Reviewer iJup for the valuable suggestions from Reviewer iJup. To assess the performance of Model Spider in other modalities, such as natural language processing, we followed the approach outlined in LogME. **We introduced cased BERT-D, uncased BERT-D, RoBERTa, and RoBERTa-D as pre-trained models to evaluate the performance of Model Spider**. We conducted our evaluation on the MRPC and SST-2 downstream datasets. **The comprehensive results are presented in Table 2 of the general response, highlighting Model Spider's consistent and robust generalization capabilities**. We will include these additional experimental results in the final version for further clarification.
>
> Additionally, we have showcased the generalization results on large language models in Table 3 of the general response. These results further highlight that Model Spider maintains strong scalability and exceptional performance across diverse tasks.
>
> &nbsp;
>
> **Question 2**: Whether the proposed method can handle larger models, such as ViT, BERT, and GPT. It would be more exciting if larger models (especially large language models) could be easily evaluated.
>
> **Answer 2**: Thank Reviewer iJup for the forward-looking feedback. We have indeed addressed the suggestions comprehensively. As described in Answer 1, **we incorporated the ViT-B/16 [4] into our existing model zoo**, covering larger models. Moreover, in alignment with the inquiry about NLP pre-trained models, **we evaluated the model ranking capabilities on the BERT series**. These results are presented in Table 1 of the general response. Additionally, in Table 3 of the general response, **we showcased Model Spider's performance on GPT-type large language models**, reaffirming its remarkable extensibility and outstanding performance.
>
> | Method | Aircraft | Caltech101 | Cars | CIFAR10 | CIFAR100 | DTD | Pets | SUN397 | Mean |
> |-------|-------|-------|-------|-------|-------|-------|-------|-------|--------|
> | NCE | 0.523 | **0.681** | **0.790** | 0.701 | 0.659 | 0.305 | 0.681 | 0.762 | 0.638 |
> | LEEP | 0.318 | 0.107 | 0.682 | 0.591 | 0.660 | 0.114 | 0.514 | 0.486 | 0.434 |
> | Model Spider | **0.693** | 0.679 | 0.781 | **0.879** | **0.955**  | **0.699** | **0.812** | **0.869** | **0.796** |
>
> **Table 1**: Comparison of NCE, LEEP, and Model Spider performance after extending the pre-trained model zoo in the original Table 1, including the addition of the ViT-B/16 pre-trained model. The remaining experimental setup is consistent with Table 1 in the paper.
>
> | Method | MRPC | SST-2 |
> |-------|-------|-------|
> | LogME | 0.493 | 1.000 |
> | Model Spider | **0.654** | 1.000 |
>
> **Table 2**: Ranking performance of pre-trained model ranking on NLP tasks.
>
> | Method | Operating System | Computer Architecture | College Physics | College Chemistry | Electrical Engineer | Metrology Engineer | Advanced Mathematics | Probability and Statistics | Modern Chinese History | Legal Professional |
> |-------|-------|-------|-------|-------|-------|-------|-------|-------|-------|-------|
> | Random | 0.083 | -0.237 | -0.185 | 0.021 | -0.532 | 0.181 | -0.010 | 0.013 | -0.005 | -0.605 |
> | ChatGPT-Top1 | 0.136 | 0.052 | 0.024 | -0.162 | 0.232 | -0.061 | -0.198 | 0.172 | 0.105 | 0.305 |
> | Model Spider | **0.720** | **0.682** | **0.311** | **0.686** | **0.308** | **0.682**  | **0.184**  | **0.243** | **0.891** | **0.737** |
>
> **Table 3**: Ranking performance of pre-trained model ranking on GPT-type LLMs, measured by weighted $\tau_w$. The horizontal axis represents evaluation datasets from C-Eval benchmark.
>
> **For more details, please see the general response.**
>
> &nbsp;
>
> **Question 3**: It would be better to consider using some methods (e.g., meta-learning) to generate initial task tokens so that new tasks can be handled without additional training samples.
>
> **Answer 3**: Thank Reviewer iJup for the sincere reply. Our approach primarily revolves around learning to rank, wherein the model's efficient ranking capability is acquired by learning from its historical performance across a diverse range of tasks. **Our learning process also shares connections with the Few-Shot Learning (FSL) scenario and metric-based approach of Meta-Learning**.
>
> Model Spider starts its journey from the performance achieved through fine-tuning established models. It then advances its capabilities through training on additional tasks, **subsequently enabling the generalization to tasks unseen before**. This is exemplified in **the cross-task setting depicted in line 267, Table 1, and Figure 3 of the paper**. Model Spider's ability to sustain its ranking across tasks of varying types and domains is notable. This capability extends **beyond the generalization power limited to a singular task type and aligns with the notion of FSL within the meta-learning framework**. This involves acquiring recognition abilities from specific subsets of categories in FSL and extending these abilities to categories that have not been encountered previously. We intend to incorporate further relevant descriptions of this aspect in the final version of our work.
>
> &nbsp;
>
> **Question 4**: The term "token" is a little confusing since the meaning of token in this paper differs from the common meaning of tokens in PTMs.
>
> **Answer 4**: Thank Reviewer iJup for the valuable feedback. We appreciate the suggestion, and in the final version, **we will use the term "model representation" or a more appropriate phrase instead of "token" to avoid confusion and ensure clarity**.
>
> Once again, **we are grateful for Reviewer iJup's time and effort in reviewing our paper**. Thank Reviewer iJup for the continued support and engagement in advancing the field.

---

### Official Review · Reviewer_PpFQ · 2023-07-05

**Soundness:** 3 good
**Presentation:** 4 excellent
**Contribution:** 4 excellent
**Rating:** 7
**Confidence:** 2

**Summary:**

This paper proposes a method to select the "best" pre-trained model for a given task. This problem is important given the large number of available pre-trained models. The key behind Spider relies on tokenizing both the models and the tasks by summarizing their characteristics into vectors. More specifically, the authors use a general encoder and measure the similarity between tokens in a supervised manner: the ranking of models are obtained through some historical tasks. Normally, one would take a list of pre-trained model, (optionally) freeze the feature-extractor part, add a randomly-initialized head on top, fine-tune on the dataset, and then measure their transferability. This process is computationally intensive.

The proposed model Spider first randomly sample training tasks and assume that we can compute the transferability for M pre-trained models and thus, their ranking. Given this dataset, the model is trained to learn a similarity function to mimic this ranking. The only features that are used are task tokens and tokens from the models. A model token consists of a representation to reflect how good the pre-trained model is in general. A task token is an embedding that represent a class in a dataset. Finally, different re-ranking strategies are proposed with efficiency-accuracy trade-off.

Transferability assessment is not my domain. Nevertheless, the proposed method seems sound to me. The experiment section is really complete and includes many datasets and baselines for single-source and multi-source model zoo. Finally, the ablation study emphasizes the importance of RankAgg.

Overall, the paper is well written, the approach seems novel, and again, given that this topic is not my domain, I don't see any reason to reject this paper.

**Strengths:**

- Efficient model selection method for a given task
- The method is novel
- Good performance in the experiment section

**Weaknesses:**

- Some concrete examples of Task & Model tokens would be appreciated

**Questions:**

I have no question

**Limitations:**

There is not a limitation section.

---

> ### Author Rebuttal · Authors · 2023-08-10
>
> **We sincerely value the insightful suggestions** provided by Reviewer PpFQ. **Within our General Response PDF file**, we have provided **concrete examples of model-task tokens**, demonstrating with pre-trained models like Food, SUN397, Caltech101, and Dogs datasets. This presentation effectively underscores the semantic relationships between various models and tasks. We are fully committed to meticulous refining and seamlessly **incorporating these details into the final version**.
>
> Once again, we extend our heartfelt and profound gratitude to Reviewer PpFQ. **Thank you very much**.

---

> > ### Comment · Reviewer_PpFQ · 2023-08-14
> >
> > Thank you for your rebuttal. I am satisfied with the answer.

---

> > > ### Author Response · Authors · 2023-08-14
> > >
> > > We sincerely appreciate Reviewer PpFQ for dedicating valuable time to offer constructive insights. Thank you very much.

---

### Official Review · Reviewer_Gb1N · 2023-07-25

**Soundness:** 3 good
**Presentation:** 3 good
**Contribution:** 3 good
**Rating:** 7
**Confidence:** 3

**Summary:**

The paper proposes a method called MODEL SPIDER for selecting the most suitable pre-trained models (PTMs) for a given downstream task. The proposed method aims to maintain a balance between efficiency and accuracy in the selection of PTMs. To achieve this, the authors tokenize all PTMs and tasks into vector representations that capture their general properties and their relationship with each other. During the training process, they dynamically select a partial set of PTMs and incorporate the specific tokens into the sampled tasks. During deployment, the authors employ a coarse-grained PTM search to narrow down the candidate PTMs and then fine-tune the selected PTMs for downstream use.

The proposed approach is evaluated on several benchmark datasets, and the results demonstrate that it outperforms existing PTM selection methods in terms of efficiency and accuracy. As part of the analysis of the effectiveness of the proposed method, the authors conduct ablation studies. They find that incorporating PTM-specific features and prompts improves the performance of the proposed method significantly.

**Strengths:**

* This paper presents a clear and well-motivated problem statement in the Introduction: how to select the best pre-trained model (PTM) for a given task. The approach section introduces the necessary preliminary and explains almost parts of the proposed method in detail.

* The method is novel and interesting, as it constructs tokens for both the PTMs and the target task, and then measures their similarity to find the optimal match. This way, the method can leverage the rich information encoded in the PTMs and adapt it to different tasks.

* The evaluation is comprehensive and detailed, covering 10 PTMs from five architectures and 9 downstream datasets for classification and regression tasks. The paper also compares the method to several strong baselines and shows that it outperforms them in selecting the most suitable PTM for each task

**Weaknesses:**

* The fitness function is a neural network that maps the PTM and task tokens to a scalar score, but the paper does not specify how this score is calculated or which threshold is compared with.

* The paper lacks details on the design and training of the task encoder, which is a crucial component of the method. In line 42-43, the authors mentioned that they use a Transformer module and refer to the reference [72] "Attention is all you need". In my opinion, this is an important detail to explain how the tokenization process is performed.

* The paper does not explain how the authors handle noise and irrelevant data in the task tokenization process."

**Questions:**

Do we need to represent all data instances, or can we use a sampling strategy to select the most representative examples from the data for encoding?

**Limitations:**

The paper does not include a limitation section that explicitly discusses the drawbacks or challenges of the proposed approach.

---

> ### Author Rebuttal · Authors · 2023-08-10
>
> **Thank Reviewer Gb1N for the insightful review** and for recognizing the strengths of our paper and the Model Spider method for pre-trained model selection. **We're grateful for Reviewer Gb1N's recognition of our novel approach**, which leverages tokens representing both pre-trained models and tasks to measure their similarity.
>
> **We responded to reviewer Gb1N's inquiries as follows:**
>
>
> **Question 1**: How this score $\hat{\mathrm{t}}_{\phi_m \rightarrow \mathcal{T}}$ is calculated and dose it need a threshold.
>
> **Answer 1**: Thank Reviewer Gb1N for the insightful feedback. **We calculate the model-task score using Equation 6, as outlined in lines 188-190.** Specifically, we employ **a transformer-based module** to compute the similarity between model and task tokens. This is achieved through a one-layer Transformer, a self-attention mechanism that accommodates various inputs, involving multi-head self-attention, multi-layer perceptron, and layer norm blocks in alternating layers. **The input to the Transformer consists of a union set of model and task tokens** denoted as
>
> $\boldsymbol{z}=\left[\boldsymbol{\theta}_{\boldsymbol{m}}, \boldsymbol{\mu}(\mathcal{T})\right] \in \mathbb{R}^{d \times(1+C)}$,
>
> leading to the similarity score $\hat{\mathrm{t}}_{\phi_m \rightarrow \mathcal{T}}$ computed as:
>
> $\operatorname{sim}\left(\boldsymbol{\theta}_m, \boldsymbol{\mu}(\mathcal{T})\right)=\mathrm{FC}(\operatorname{transformer}(\boldsymbol{z})[0])$.
>
> For downstream tasks, we calculate scores for multiple pre-trained models, creating a ranking based on these scores. **Our focus is on the ranking order, where higher scores correspond to better downstream performance.** (lines 118-121). The calculation process **does not involve a threshold** for comparison. **The weighted** $\tau_w$ metric evaluates the quality of rankings **based on relative order rather than absolute scores**. The $\tau_w$ considers differences in ranking positions and is not concerned with specific score. **We will provide more details in the final version**.
>
> &nbsp;
>
> **Question 2**: What are the construction details of the task encoder.
>
> **Answer 2**: Thank Reviewer Gb1N for the perceptive feedback. The method's task encoder is indeed a critical component. **The task representation (tokens) is derived using an additional self-supervised training tokenizer, denoted as** $\psi$, **with relevant explanations in the Task Token section (lines 177-179, 272).** This encoder is an additional frozen unit **with the same parameter magnitudes as the pre-trained models to be ranked**.
>
> We detail in line 272 (or in Appendix line 35) that $\psi$ **is realized through a pre-trained Swin-B-based EsViT** [1,2] (accessible at https://github.com/microsoft/esvit), trained on ImageNet-1K using self-supervised learning. In our experiments, this encoder functions as a feature extractor. **Additionally, the Transformer-based module (in lines 42-43) details for assessing model-task similarity in Model Spider** are expounded upon in lines 186-188 and response to the previous question. We will further augment this in the final version.
>
> [1] Efficient self-supervised vision transformers for representation learning.
> [2] Swin transformer: Hierarchical vision transformer using shifted windows.
>
> &nbsp;
>
> **Question 3**: How to handle noise and irrelevant data in the task tokenization process.
>
> **Answer 3**: Thank Reviewer Gb1N for the thoughtful inquiry. **While our task tokenizer might include noise and irrelevant data, Model Spider effectively mitigates noise effects on task representations.** We sample tasks from a mixed dataset during training (lines 273, 293, and Appendix line 149), enhancing diversity. **Model Spider adeptly captures task information,** utilizing numerous training tasks (Appendix line 153) and random sampling to ensure task variability, minimizing noise impact.
>
> Moreover, in the experiments presented **in the latter part of Table 1**, we assess the performance of pre-trained model ranking with employing a few-shot manner (10 examples per class), and each result is repeated 30 times for evaluation. **In few-shot tasks, where the sample size is limited, noise and irrelevant data have a more pronounced impact on task representations (tokens).** Nonetheless, Model Spider demonstrates stable and superior performance even under these conditions. We appreciate Reviewer Gb1N's consideration of these aspects within our methodology.
>
> &nbsp;
>
> **Question 4**: Do we need to represent all data instances, or use a sampling strategy for encoding.
>
> **Answer 4**: Thank Reviewer Gb1N for the precise question. In Model Spider, **we employ random sampling** to create task tokens by selecting 50 instances per class, elaborated in Appendix, line 153. **Our method does not necessitate representing every instance or adopting a sampling strategy to choose the most representative examples.** Model Spider's robustness enables effective learning from randomly sampled task representations (tokens). Our approach of random task sampling is detailed in Appendix lines 153 and 187: single-source experiment (Table 1) uses around 1k tasks, and multi-source (Figure 3) employs about 4k tasks. Ample sampling is employed to mitigate the influence of randomness during training and testing to the fullest extent possible.
>
> &nbsp;
>
> **Question 5**: The paper does not include a limitation section that explicitly discusses the drawbacks or challenges of the proposed approach.
>
> **Answer 5**: Thank Reviewer Gb1N. **Our Discussions and Limitations section can be found on line 252 of the appendix.** We will ensure its placement is appropriately adjusted in the final version.
>
> **We sincerely thank Reviewer Gb1N for their insightful review and valuable questions**. We are committed to addressing these queries thoroughly in the final version.

---

> > ### Comment · Reviewer_Gb1N · 2023-08-21
> >
> > I appreciate your efforts to address my comments and concerns. You have provided sufficient information and clarification in this rebuttal. I increased my overall rating for this paper from Borderline Accept to Accept.

---

> > > ### Author Response · Authors · 2023-08-22
> > >
> > > We sincerely appreciate Reviewer Gb1N's insights and feedback. We remain committed to further improvement. Thank you very much.

---

### Author Rebuttal · Authors · 2023-08-10

**Dear Reviewers:**

We would like to express our sincere gratitude to Reviewers Gb1N, PpFQ, iJup, 1u8a, and mKeC for **their insightful reviews of our submission**. We are heartened by the constructive feedback and valuable suggestions each reviewer provides. We acknowledge that **all reviewers** have highlighted the motivating factors behind our proposed method, Model Spider, emphasizing its **novelty and interest**. They have also recognized **our comprehensive and detailed evaluation**, as well as the **clarity and organization of our paper's presentation**.

We are pleased by the positive feedback from Reviewers Gb1N, PpFQ, iJup, and 1u8a regarding Model Spider's **outstanding performance**. We understand the concern raised by Reviewer mKeC about **future task generalization** and the questions posed by Reviewers mKeC, iJup, and 1u8a about the applicability, such as ranking on tasks with new types, modalities, or on larger pre-trained vision or language models. **To address these, we conducted extensive experiments to show Model Spider's robustness across various scenarios**.

Moving forward, we will present our supplementary experiments.

+ **For Pre-trained Larger Vision Models**

For ranking on larger vision pre-trained models, we have incorporated **the ViT-B/16 model into our existing pre-trained model zoo**. This medium-to-large-scale vision model with approximately 100 million parameters was introduced by fine-tuning only the last linear layer (linear probing). We then **compared the performance of NCE, LEEP, and Model Spider** on this extended pre-trained model zoo.

| Method | Aircraft | Caltech101 | Cars | CIFAR10 | CIFAR100 | DTD | Pets | SUN397 | Mean |
|-------|-------|-------|-------|-------|-------|-------|-------|-------|--------|
| NCE | 0.523 | **0.681** | **0.790** | 0.701 | 0.659 | 0.305 | 0.681 | 0.762 | 0.638 |
| LEEP | 0.318 | 0.107 | 0.682 | 0.591 | 0.660 | 0.114 | 0.514 | 0.486 | 0.434 |
| Model Spider | **0.693** | 0.679 | 0.781 | **0.879** | **0.955**  | **0.699** | **0.812** | **0.869** | **0.796** |

**Table 1**: Comparison of NCE, LEEP, and Model Spider performance after extending the pre-trained model zoo in the original Table 1, including the addition of the ViT-B/16 pre-trained model. The remaining experimental setup is consistent with Table 1 in the paper.

+ **For Pre-trained Models of NLP**

To assess the adaptability of Model Spider to tasks in other modalities, such as Natural Language Processing (NLP). **We introduced cased BERT-D, uncased BERT-D, RoBERTa, and RoBERTa-D as pre-trained models** to evaluate the performance of Model Spider. Our evaluation included the **MRPC** [1] and **SST-2** [2] downstream datasets. MRPC consists of sentence pairs extracted from online news sources, where the task involves determining the semantic equivalence of sentences within each pair. On the other hand, SST-2 comprises sentences from movie reviews. The task is focused on predicting the sentiment (positive/negative) of a given sentence. **This model ranking task in the field of NLP follows the approach established by LogME** [3]. We got the following results:

| Method | MRPC | SST-2 |
|-------|-------|-------|
| LogME | 0.493 | 1.000 |
| Model Spider | **0.654** | 1.000 |

**Table 2**: Ranking performance of pre-trained model ranking on NLP tasks.

+ **For Pre-trained Larger Language Models**

Furthermore, we perform pre-trained model ranking for Large Language Models (LLM) of GPT-type, which includes ChatGPT, ChatGLM2-6B, Qwen-7B, Baichuan-7B, MOSS, bloomz-mt-176B, and Chinese Alpaca-13B. This involves **introducing the C-Eval Benchmark** [4] and employing Model Spider to rank these models' performance **across 10 sub-evaluation datasets encompassing domains** as science, technology, engineering, mathematics, and humanities. **We utilize embeddings available at** https://huggingface.co/GanymedeNil/text2vec-large-chinese **to encode the downstream tasks**. We leverage historical performance from the other datasets provided by C-Eval as the training set for Model Spider, with the objective of learning model and task representations (tokens). As the output access interface of LLMs is limited (for instance, obtaining features from ChatGPT is challenging), common methods fail in such scenarios. We compare our model ranking approach **against these two setups**: a random ranking, and ranking ChatGPT as the top model with others in random order. The obtained results are as follows:

| Method | Operating System | Computer Architecture | College Physics | College Chemistry | Electrical Engineer | Metrology Engineer | Advanced Mathematics | Probability and Statistics | Modern Chinese History | Legal Professional |
|-------|-------|-------|-------|-------|-------|-------|-------|-------|-------|-------|
| Random | 0.083 | -0.237 | -0.185 | 0.021 | -0.532 | 0.181 | -0.010 | 0.013 | -0.005 | -0.605 |
| ChatGPT-Top1 | 0.136 | 0.052 | 0.024 | -0.162 | 0.232 | -0.061 | -0.198 | 0.172 | 0.105 | 0.305 |
| Model Spider | **0.720** | **0.682** | **0.311** | **0.686** | **0.308** | **0.682**  | **0.184**  | **0.243** | **0.891** | **0.737** |

**Table 3**: Ranking performance of pre-trained model ranking on GPT-type LLMs, measured by weighted $\tau_w$. The horizontal axis represents evaluation datasets from C-Eval benchmark.

**Thank you for your consideration and support.** We are committed to addressing the reviewers' feedback and further improving our work based on the insightful comments. **We once again extend our appreciation to the reviewers for their invaluable contributions**.

[1] Automatically constructing a corpus of sentential paraphrases.

[2] Recursive deep models for semantic compositionality over a sentiment treebank.

[3] Logme: Practical assessment of pre-trained models for transfer learning.

[4] C-eval: A multi-level multi-discipline chinese evaluation suite for foundation models, https://cevalbenchmark.com/static/leaderboard.html

---

### Decision · Program_Chairs · 2023-09-21

**Decision:**

Accept (spotlight)

**Comment:**

This is a well-written paper that proposes a very interesting and novel approach for automatically selecting the best pre-trained model for a given task. This is a straightforward but effective approach that has significant practical utility. The experimental evaluation is rigorous and covers a wide range of pre-trained models, downstream tasks, etc.

The reviewers raised a few concerns about this paper, but the most critical aspects of those were adequately addressed during the rebuttal.

Given that the strengths of this paper clearly outweigh its weaknesses, this paper is suitable for publication.

The authors are strongly encouraged to carefully consider all of the reviewer feedback, including the rebuttal-related discussion, and to take meaningful steps to incorporate it into the final version of their paper.